



# Plant phenology evaluation of CRESCENDO land surface models. Part I: start and end of growing season.

Daniele Peano[1], Deborah Hemming[2], Stefano Materia[1], Christine Delire[3], Yuanchao Fan[4,5], Emilie Joetzjer[3], Hanna Lee[4], Julia E.M.S. Nabel[6], Taejin Park[7,8], Philippe Peylin[9], David Wårlind[10], Andy Wiltshire[2,11], and Sönke Zaehle[12]

[1]Fondazione Centro Euro-Mediterraneo sui Cambiamenti Climatici, CSP, Bologna, Italy
[2]Met Office Hadley Centre, Exeter, UK
[3]Centre National de Recherches Météorologiques, UMR3589, Université de Toulouse/Météo-France/CNRS, Toulouse, France
[4]NORCE Norwegian Research Centre AS, Bjerknes Centre for Climate Research, Bergen, Norway
[5]Harvard University, Cambridge, USA
[6]Max Planck Institute for Meteorology, Hamburg, Germany
[7]NASA Ames Research Centre, CA, USA
[8]Bay Area Environmental Research Institute, CA, USA
[9]Laboratoire des Sciences du Climat et l'Environnement, Gif-sur-Yvette, France
[10]Department of Physical Geography and Ecosystem Science, Faculty of Science, Lund University, Sweden
[11]Global Systemss Insitute, University of Exeter, Exeter, UK
[12]Max Planck Institute for Biogeochemistry, Jena, Germany

**Correspondence:** Daniele Peano (daniele.peano@cmcc.it)

**Abstract.** Plant phenology plays a fundamental role in land-atmosphere interactions, and its variability and variations are an indicator of climate and environmental changes. For this reason, current land surface models include phenology parameterizations and related biophysical and biogeochemical processes. In this work, the climatology of beginning and end of the growing season, simulated by seven state-of-the-art European land surface models, is evaluated globally against satellite observations. The assessment is performed using the vegetation metric leaf area index and a recently-developed approach, named four growing season types. On average, the land surface models show a 0.6-month delay in the growing season start, while they are about 0.5 months earlier in the growing season end. Difference with observation tends to be higher in the Southern Hemisphere compared to the Northern Hemisphere. High agreement between land surface models and observations is exhibited in areas dominated by broad-leaf deciduous trees, while high variability is noted in regions dominated by broad-leaf deciduous shrubs. Generally, the timing of the growing season end is accurately simulated in about 25% of global land grid points versus 16% in the timing of growing season start. The refinement of phenology parameterization can lead to better representation of vegetation-related energy, water, and carbon cycles in land surface models, but plant phenology is also affected by plant physiology and soil hydrology processes. Consequently, phenology representation and, in general, vegetation modelling is a complex task, which still needs further improvement, evaluation, and multi-model comparison.





# 1 Introduction

Plant phenology and its variability have a substantial influence on the terrestrial ecosystem (e.g. *Churkina et al.*, 2005; *Kucharik et al.*, 2006; *Berdanier and Klein*, 2011) and land-atmosphere interactions (e.g. *Cleland et al.*, 2007; *Richardson et al.*, 2013; *Keenan et al.*, 2014) making phenology variability one of the indicators of climate change (e.g. *Schwartz et al.*, 2006; *Soudani et al.*, 2008; *Jeong et al.*, 2011) since modifications in both spring and autumn phenology are reported in recent decades under

global warming (e.g. *Menzel et al.*, 2006; *Richardson et al.*, 2013; *Zhu et al.*, 2016; *Chen et al.*, 2020; *Zhang et al.*, 2020).

Given the influence of plant phenology on vegetation productivity, and since green leaves are the primary interface for the exchange of energy, mass (e.g., water, nutrient, and $CO_2$), and momentum between the terrestrial surface and the planetary boundary layer (*Richardson et al.*, 2012), land surface models (LSMs) need to accurately simulate plant growing season cycles. Limitations may result in biases and uncertainties in representing vegetation productivity and carbon cycle (e.g. *Churkina et al.*,

2005; *Kucharik et al.*, 2006; *Berdanier and Klein*, 2011; *Richardson et al.*, 2012; *Friedlingstein et al.*, 2014; *Savoy and Mackay*, 2015; *Buermann et al.*, 2018). For example, *Kucharik et al.* (2006) show an overestimated April-May net ecosystem production triggered by biases in plant budburst. *Berdanier and Klein* (2011) describe a link between above ground net primary production, growing season length, and soil moisture in high-elevation meadows. They show that the potential impact of changes in active growing season length on biomass production accounts for about 3-4 g m$^{-2}$ d$^{-1}$. The work by *Richardson et al.* (2012) is

an example of a systematic evaluation of LSMs' phenology representation. They evaluate fourteen models participating in the North American Carbon Program Site Synthesis against ten forested sites, within the AmeriFlux and Fluxnet-Canada networks. Their assessment reveals a typical bias of about two weeks in LSMs representation of the beginning and end of the growing season. They also show a low skill in LSMs' reproduction of the observed inter-annual phenology variability. These biases lead to an overestimation of about 235 gC m$^{-2}$ yr$^{-1}$ in the gross ecosystem photosynthesis of deciduous forest sites. However,

uncertainties in simulated maximum production partially balance this overestimation. The work by *Buermann et al.* (2018) is another example of a multi-LSMs evaluation. They observe widespread lagged plant productivity responses across northern ecosystems associated with warmer and earlier springs, which is weakly captured by ten evaluated TRENDYv6 current LSMs. Consequently, current LSMs still present biases in simulating timings and the magnitude of the vegetation active season.

The latest generation of LSMs have started including a more detailed description of land biophysical and biogeochemical

processes, and they have become able to explicitly represent carbon and nitrogen land-cycles, as well as plant phenology and related water and energy cycling on a global scale (e.g. *Oleson et al.*, 2013; *Lawrence et al.*, 2018). In particular, current LSMs link Leaf Area Index (LAI) and plant phenology to changes in temperature, precipitation, soil moisture and light availability (e.g. *Oleson et al.*, 2013; *Lawrence et al.*, 2018), as displayed in observations (e.g. *Caldararu et al.*, 2012; *Zeng et al.*, 2013; *Tang and Dubayah*, 2017).

In this framework, the European CRESCENDO project (https://www.crescendoproject.eu/) fostered the development of a new generation of LSMs to be used as the land component of the Earth System Models (e.g. *Smith et al.*, 2014; *Olin et al.*, 2015; *Cherchi et al.*, 2019; *Mauritsen et al.*, 2019; *Sellar et al.*, 2020; *Seland et al.*, 2020; *Yool et al.*, 2020; *Boucher et al.*, 2020) employed in the Coupled Model Intercomparison Project Phase 6 (CMIP6, *Eyring et al.*, 2016). In particular, seven novel





LSMs, which are part of the CRESCENDO effort, are used in this work, namely Community Land Model (CLM) version 4.5
(*Oleson et al.*, 2013) and version 5.0 (*Lawrence et al.*, 2019), JULES-ES (*Wiltshire et al.*, 2020), JSBACH (*Mauritsen et al.*,
2019), LPJ-GUESS (*Lindeskog et al.*, 2013; *Smith et al.*, 2014; *Olin et al.*, 2015), ORCHIDEE (*Krinner et al.*, 2005), and
ISBA-CTRIP (*Decharme et al.*, 2019).

Given the relevance of plant phenology and its changing variability related to climate, LSMs need routine evaluation against
observations (e.g. *Richardson et al.*, 2012; *Dalmonech and Zaehle*, 2013; *Murray-Tortarolo et al.*, 2013; *Anav et al.*, 2013;
*Peano et al.*, 2019). This study aims to evaluate the ability and limits of the novel CRESCENDO LSMs to represent the
global climatology of start and end of growing season timings. The CRESCENDO LSMs cover a wide range of phenology
schemes and vegetation descriptions. This selection may therefore help understand the sources of differences between LSMs'
representation of phenology and the regions where plant phenology simulations remain difficult.

The novel Four Growing Season Types (4GST) methodology developed by *Peano et al.* (2019) is used to evaluate phenology
based on LAI data. Most ecosystem and climate models introduce 'leaf area' as a fundamental state parameter describing the
interactions between the biosphere and the atmosphere. The most common measure of the area of leaves is the LAI, which is
generally defined as the one-sided leaf surface area divided by the ground area in $m^2/m^2$ (*Chen and Black*, 1992). In addition,
LAI is the key variable by which LSMs scale-up leaf-level processes to canopy and ecosystem scale exchanges of carbon,
energy, and water. This makes the LAI a reasonable choice for the evaluation of the LSMs' phenology (*Murray-Tortarolo*
*et al.*, 2013; *Peano et al.*, 2019).

In this paper, we present a brief description of the methods, LSMs and satellite data used (section 2). Next, we present the
main results of the satellite data comparison and evaluation of LSMs against observations (section 3). Finally, we discuss the
methodology, data, and results (section 4), followed by concluding remarks (section 5).

## 2  Method, models and data

### 2.1  Satellite observation

To perform a comprehensive global phenology evaluation, a satellite-based observational dataset is required. LAI satellite
observations present uncertainties and limitations related to the assumptions and algorithms applied in the LAI calculation
(Section 4.2 e.g. *Fang et al.*, 2013; *Jiang et al.*, 2017). For this reason, three satellite observational products are considered in
this work, which differ in acquisition sensor (i.e. AVHRR for LAI3g, MODIS for MODIS LAI, and SPOT/PROBA VEGETA-
TION for SENTINEL; *Piao et al.*, 2020) and methodologies for data-acquisition: LAI3g (*Zhu et al.*, 2013), MODIS version 6
(https://lpdaac.usgs.gov/ ; *Myneni*, 2015a, b), and SENTINEL version 2 (https://www.copernicus.eu/ ; *Drusch et al.*, 2012). The
full time series of LAI3g data is generated by an artificial neural network (ANN) algorithm that is trained with the overlapping
data of NDVI3g and Terra Moderate Resolution Imaging Spectroradiometer (MODIS) LAI products (see *Zhu et al.*, 2013). It
covers the 1982-2011 period with a 15-day temporal frequency and a $1/12°$ spatial resolution. The MODIS LAI algorithm is
based on a three-dimensional radiative transfer equation that links surface spectral bi-directional reflectance factors to vegeta-
tion canopy structural parameters (see *Yan et al.*, 2016a). It covers the 2000-2017 period with a 4- or 8-day temporal frequency





and a 500 m spatial resolution. SENTINEL LAI dataset is obtained through a neural network applied on top-of-atmosphere input reflectances in red and near-infrared bands derived from PROBA-V. The instantaneous LAI estimates obtained in this way go through a temporal smoothing and small gap filling, which discriminate between evergreen broadleaf forest and no-
evergreen broadleaf forest pixels (see *Verger et al.*, 2019). It covers the 1999-2019 period with a 10-day temporal frequency and a 1km spatial resolution. Note that SENTINEL has a reduced latitudinal cover compared to MODIS and LAI3g since it covers up to 75°N versus the 90°N of the other two products.

The 2000-2011 period is common to the three satellite datasets and it is used in the present analysis. The satellite observations are aggregated into monthly values and regridded, by means of a first order conservative remapping scheme (*Jones*, 1999), to
a regular 0.5° x 0.5° grid for consistency with the LSMs' output.

To assess the model phenology evolution at regional scale, the observed ESA CCI land cover map has been used (https://www.esa-landcover-cci.org/). In particular, *Li et al.* (2018) aggregated the original 37 ESA-CCI land cover classes into half-degree spatial resolution and translate them into 14 Plant Functional Types (PFTs) based on an adjusted cross-walking table. These data have been used to obtain an observed dominant PFT map for the 2000-2011 period. Based on *Li et al.* (2018), all vegetation types are
classified into ten categories: Broad-leaf Evergreen Trees (BET); Broad-leaf Deciduous Trees (BDT); Needle-leaf Evergreen Trees (NET); Needle-leaf Deciduous Trees (NDT); Broad-leaf Evergreen Shrubs (BES); Broad-leaf Deciduous Shrubs (BDS); Needle-leaf Evergreen Shrubs (NES); Needle-leaf Deciduous Shrubs (NDS); grass covered areas (Grass); crop covered areas (Crop).

## 2.2 Land surface models

Seven European LSMs, which are part of the CRESCENDO project, are evaluated in this study. Further details on each of these LSMs are provided in the following sections, and briefly summarized in Table 1.

### 2.2.1 Community Land Model version 4.5

The Community Land Model (CLM) is the terrestrial component of the Community Earth System Model version 1.2 (CESM1.2, http://www.cesm.ucar.edu/models/cesm1.2/), and, in its version 4.5 (CLM4.5, *Oleson et al.*, 2013) and biogeochemical con-
figuration (i.e. BGC compset, *Koven et al.*, 2013), it is the land component of the CMCC coupled model version 2 (CMCC-CM2, *Cherchi et al.*, 2019). CLM4.5-BGC features fifteen PFTs, in which crop is represented as a generic C3 crop. The PFTs time-evolution follow the area changes described in the Land Use Harmonization version 2 (LUH2, *Hurtt et al.*, 2020). CLM4.5-BGC resolves explicitly the carbon-nitrogen bio-geochemical cycles (*Oleson et al.*, 2013; *Koven et al.*, 2013), including plant phenology, which is described by means of three specific parameterization: (1) evergreen plant phenology; (2)
seasonal-deciduous plant phenology; (3) stress-deciduous plant phenology (*Oleson et al.*, 2013).

The evergreen plant phenology is characterized by a background litterfall, that is a continuous leaf fall and fine roots turnover distributed along the year. A PFT-specific leaf longevity parameter drives this mechanism (*Oleson et al.*, 2013).

The seasonal-deciduous plant phenology derives from the model Biome-BGC version 4.1.2 (*Thornton et al.*, 2002) and parameterizations by *White et al.* (1997). The leaf onset starts when the soil temperature of accumulated growing-degree-day



(GDD) passes a critical threshold. The leaf litterfall, instead, starts when the day-length exceeds another specific threshold (*Oleson et al.*, 2013). Summer and winter solstices also act as time-limiting factors.

Finally, the stress-deciduous plant phenology involves grass and trees that respond to both cold and drought stresses. This parameterization is based on the *White et al.* (1997) grass phenology model. The leaf onset is soil moisture-driven in areas characterized by year-round warm conditions, while both soil moisture and soil temperature drive the leaf onset in region

characterized by a cold season. Sustained period of dry soil or cold temperature, or day-length shorter than six hours trigger the end of the growing season. Further details can be found in *Oleson et al.* (2013).

### 2.2.2 Community Land Model version 5.0

CLM version 5.0 (CLM5.0) is the terrestrial component of the Community Earth System Model version 2 (CESM2, http://www.cesm.ucar.edu/models/cesm2/) and of the Norwegian Earth System Model version 2 (NorESM2, *Seland et al.*,

125 2020).

CLM5.0 uses the same number of default PFTs and three specific plant phenology parameterization applied in CLM4.5 (*Lawrence et al.*, 2018). Additionally, the CLM5.0 crop module uses two C3 crop configurations: C3 rainfed and C3 irrigated. The irrigation area is based on crop type and region, and the irrigation triggers for crop phenology are newly updated from the CLM4.5. New phenology-related features in CLM5.0 include additional antecedent rain requirement trigger for stress decidu-

ous phenology to reduce the occurrence of anomalous green-up during the dry season driven by upwards water movement from wet to dry soil layers (*Dahlin et al.*, 2015). In addition, several major changes have been made in the CLM5.0. One of the major physiological changes include maximum stomatal conductance, which now uses the Medlyn conductance model (*Medlyn et al.*, 2011), rather than the previously used Ball-Berry stomatal conductance model. In CLM5.0, the *Jackson et al.* (1996) rooting profiles are used for both water and carbon, where the rooting depths were increased for broadleaf evergreen and broadleaf

deciduous tropical tree PFTs. As soil moisture is an important control on plant phenology, the modified rooting profiles would affect the stress deciduous trigger and other water-related processes. Other modifications include nutrient dynamics, hydrological and snow parameterizations, plant hydraulic functions, revised nitrogen cycling with flexible leaf stoichiometry, leaf N optimization for photosynthesis, and carbon costs for plant nitrogen uptake (*Lawrence et al.*, 2019).

### 2.2.3 JULES-ES

JULES-ES is the Earth System configuration of the Joint-UK Land Environment Simulator (JULES). JULES-ES is the terrestrial component of the new UK community ESM, UKESM1 (*Sellar et al.*, 2020). It is based on the core physical land configuration of JULES (JULES-GL7) as described in *Wiltshire et al.* (2020). JULES-GL7 simulates the exchange of heat, water and momentum between the land and atmosphere using prescribed land-cover including LAI. JULES-ES simulates the exchange of carbon and the change in surface properties that control the physical interaction between the land and atmosphere

including LAI and landcover. JULES-ES includes a full carbon and nitrogen cycle with dynamic vegetation, 13 Plant Functional Types with trait based physiology (*Harper et al.*, 2016), and a representation of crop harvest and landuse change. In JULES-ES, the allometrically defined maximum LAI varies with the carbon status (*Clark et al.*, 2011) and extent of the under-





lying vegetation. In the case of natural grasses LAI can vary rapidly sub-seasonally whereas tree PFTs have a smaller variation. Phenology operates on top of this variation for Deciduous Broadleaf and Needleleaf PFTs based on an accumulated thermal
time model.

The simulations described here used a near-final configuration of JULES-ES prior to the final tuning performed as part of UKESM1 (*Yool et al.*, 2020). JULES-ES is run offline forced by global historic meteorological data as described in section 2.3.

### 2.2.4  JSBACH

JSBACH3.2, the JSBACH version used for the CRESCENDO simulations, is the land component of the MPI-ESM1.2 (*Mau-*
*ritsen et al.*, 2019). For the simulations described here, JSBACH3.2 is run offline at T63 (∼1.9°) resolution, mostly according to the common CRESCENDO protocol as described in section 2.3. Simulations were conducted without natural changes in the land cover, instead a static map of natural land cover based on *Pongratz et al.* (2008) was used. Anthropogenic land cover changes were applied using land-use transitions (see *Reick et al.*, 2013) derived from the LUH2 forcing, whereby rangelands were treated as natural vegetation (see also *Mauritsen et al.*, 2019). To avoid a cold start problem when applying land-use tran-
sitions, simulations were already started in 1700. JSBACH3.2 contains a multilayer hydrology model (*Hagemann and Stacke*, 2015) and a representation of the terrestrial nitrogen cycle (*Goll et al.*, 2017).

JSBACH3.2 is run with its default phenology model, called LoGro-P, as evaluated in *Böttcher et al.* (2016) and *Dalmonech et al.* (2015). This phenology is based on a logistic equation for the temporal development of the LAI. Under ideal environmental conditions, the LAI approaches a maximum value representing a prescribed PFT specific physiological limit. Growth
and leaf shedding rates of the logistic equation are functions of environmental conditions, chosen differently according to the phenology type (see e.g. *Böttcher et al.*, 2016). JSBACH3.2 distinguishes the following phenology types: (1) evergreen, (2) summergreen, (3) raingreen, (4) grasses, as well as (5) tropical and extra-tropical crops. (1) For the evergreen phenology, two subsequent phases are distinguished: a "growth phase" (spring), characterized by the absence of leaf shedding and positive growth, and a "rest phase" (all other seasons), with non-zero leaf shedding and no growth. (2) For the summergreen phenology
a third phase is added in between those two phases: a "vegetative phase" (summer) with no growth as in the rest phase but with a small shedding rate. The phases of both, the evergreen and the summergreen phenologies, are determined by temperature thresholds calculated by the alternating model of *Murray et al.* (1989) from heat sums, chill days and critical soil temperatures. (3) The raingreen phenology does not have distinct phases, and the growth rate is non-zero whenever the soil moisture exceeds the wilting point and the net primary productivity (NPP) is positive. The shedding rate depends on the relative soil water
content. (4) The grass phenology resembles the raingreen phenology but further requires the air temperature and soil moisture to exceed a critical value for a non-zero growth rate. Because grass roots are less deep than tree roots, soil moisture is taken from the upper soil layer in the grass phenology, while for the raingreen phenology the water content of the whole soil column is considered. (5) In contrast to the other phenology types, the growth rate of the crop phenology is modeled as a function of NPP. JSBACH distinguishes tropical and extra-tropical crops in order to reflect different farming practices in dependence
of the prevailing climatic conditions: tropical crops have no distinct phase switches, and require sufficient temperature, an upper layer soil moisture above wilting point and a positive NPP. Extra-tropical crops have distinguished phases for summer





and winter crops which strongly depend on temperature (heat sum) determining the growth and the shedding rate (emulating harvest events).

The vegetation in the conducted JSBACH3.2 simulations was represented by 12 PFTs, each of which is linked to one of the
phenology types: one forest type with evergreen phenology, one forest and one shrub type with summergreen phenology, two forest and one shrub type with raingreen phenology, C3 and C4 grasses as well as C3 and C4 pastures with grass phenology, and C3 and C4 crops with extra-tropical and tropical crop phenology, respectively.

### 2.2.5 LPJ-GUESS

The Lund-Potsdam-Jena General Ecosystem Simulator version 4.0 (LPJ-GUESS; *Lindeskog et al.*, 2013; *Smith et al.*, 2014;
*Olin et al.*, 2015), a process-based 2nd generation dynamic vegetation and biogeochemistry model, is the terrestrial biosphere component used in the European community Earth-System Model (EC-Earth-Veg, http://www.ec-earth.org/; *Hazeleger and Bintanja*, 2012; *Döscher et al.*, in prep.; *Miller et al.*, in prep.). It simulates vegetation dynamics, land use and land management following LUH2 (*Hurtt et al.*, 2020). LPJ-GUESS features 25 PFTs, ten woody and two herbaceous PFTs compete in the natural stand fractions, whereas two herbaceous species, C3 and C4 photosynthesis-pathways, compete in pasture, urban and peatland
fractions. Crop stands have each five crop functional types representing the properties of global crop types and corresponds to the classes found in LUH2, namely both annual and perennial C3 and C4 crops, and C3 N fixers, and two herbaceous cover crops (C3 and C4) that are grown in-between the main agricultural growing seasons. The interactive carbon-nitrogen biogeochemical cycles in LPJ-GUESS induces nutrient limitation on natural vegetation and crop growth, and decomposition rate of soil organic matter that will influence soil biogeochemistry, CO2 fluxes and N trace gas emissions (*Smith et al.*, 2014;
*Olin et al.*, 2015).

Plant phenology is described by means of three specific parameterization: (1) evergreen plant phenology; (2) seasonal-deciduous plant phenology; (3) stress-deciduous plant phenology (*Smith et al.*, 2014). An explicit phenological cycle is simulated only for leaves and fine roots in seasonal-deciduous and stress-deciduous PFTs, whereas evergreen PFTs have a prescribed background litterfall for leaves, fine roots and sapwood. Seasonal-deciduous plant phenology is based on a PFT-dependent ac-
cumulated GDD sum threshold for leaf onset, with leaf area rising from 0 to the pre-determined annual maximum leaf area linearly with an additional 200 (100 for herbaceous and needleleaved tree PFTs) degree days above a threshold of 5 °C. For seasonal-deciduous PFTs, growing season length is fixed, all leaves being shed after the equivalent of 210 days with full leaf cover. Stress-deciduous plant phenology PFTs shed their leaves whenever the water stress scalar $\omega$ drops below a threshold, $\omega_{\min}$, signifying the onset of a drought period or dry season. New leaves are produced, after a prescribed minimum dormancy
period, when $\omega$ again rises above $\omega_{\min}$ (*Smith et al.*, 2014). Crop PFT sowing and harvest decisions are modelled based on climate variability (*Waha et al.*, 2011; *Lindeskog et al.*, 2013) and climatic thresholds (*Bondeau et al.*, 2007).

### 2.2.6 ORCHIDEE

The ORCHIDEE model used for the CRESCENDO simulations is the land component of the IPSL (Institut Pierre Simon Laplace) ESM and the version corresponds to the one used for the CMIP6 simulations (*Boucher et al.*, 2020). The model





calculates primarily the fluxes (and stocks) of water, energy, and carbon between the different soil and plant reservoirs and the fluxes with the atmosphere. Since its first description in *Krinner et al.* (2005), the model has substantially evolved; we describe below only the main features relevant for this study.

The surface heterogeneity is described with 15 different PFTs that can be mixed in each grid cell. The annual evolution of the PFT distribution is derived from the LUH2 database as described in more details in *Lurton et al.* (2019). All PFTs

share the same equations but with different parameters, except for the leaf phenology. They are grouped into three soil tiles, with independent hydrological budget, according to their physiological behavior: high vegetation (forests) with eight PFTs, low vegetation (grasses and crops) with 6 PFTs, and bare soil with one PFT. Only one energy budget and one snow budget is calculated for the whole grid cell.

For the carbon cycle, photosynthesis is parameterized based on *Farquhar et al.* (1980) and *Collatz et al.* (1992) for C3 and

C4 plants, respectively, using the implementation proposed by *Yin et al.* (2009). Once the carbon is fixed by photosynthesis, growth and maintenance respirations are calculated and the remaining carbon is then allocated into 8 plant compartments (below and above ground sapwood and heartwood; leaves; fruit; roots; reserves), following the so-called resource limitations approach (*Friedlingstein et al.*, 1999). Each compartment has a specific turnover and the dead biomass enters to the soil via four litter pools, which then feeds three soil organic carbon pools, following the CENTURY model (*Parton et al.*, 1987).

A Phenology module describes leaf onset and leaf senescence for deciduous PFTs. In temperate and boreal regions, leaf onset is driven by an accumulation of warm temperature in spring, following the concept of Growing Degree Days (GDD). In addition a minimum period of cold temperature, based on a Number of Chilling Days (NCD), is used to avoid buds dying with late frosts. Both criteria are combined, with PFT-specific GDD and NCD thresholds to be met, before leaves can start growing. For the dry tropics and semi-arid ecosystems, a moisture availability criteria is used based on water accumulated

in soil. Both temperature and moisture criteria are combined for grasses and crops and the different parameters of the leaf onset parameterisation have been calibrated with satellite data (*Botta et al.*, 2000). Leaves are then further separated into four age classes with different photosynthetic efficiency. Leaf fall is controlled by different turnover processes. The first one is a simple aging process and a second senescence process based on climatic conditions (either based on air temperature or on soil moisture conditions) is applied.

**2.2.7  ISBA-CTRIP**

ISBA-CTRIP is the land surface model of CNRM-ESM2-1 (http://www.umr-cnrm.fr/spip.php?article1092). It is used within the SURFEX version 8 modeling platform representing SURFace EXchanges between ocean, lakes, and land. It solves the energy, carbon and water budgets at the land surface and was recently thoroughly upgraded.

*Decharme et al.* (2019) give a detailed description of the physical aspects of the model. ISBA-CTRIP simulates plant

physiology (photosynthesis and autotrophic respiration), carbon allocation and turnover, and carbon cycling through litter and soil. It includes a module for wild fires, land cover changes, and carbon leaching through the soil and transport of dissolved organic carbon to the ocean. Leaf photosynthesis is represented by the semi-empirical model proposed by *Goudriaan et al.* (1985). Canopy level assimilation is calculated using a 10-layer radiative transfer scheme including direct and diffuse radiation.





Vegetation in ISBA-CTRIP is represented by 4 carbon pools for grasses and crops (leaves, stem, roots and a non-structural
carbohydrate storage pool) with 2 additional pools for trees (aboveground wood and coarse roots). Leaf phenology results
directly from the daily carbon balance of the leaves. Leaf turnover time is dependent on potential leaf longevity reduced when
10-day assimilation rates start to decrease. Leaf area index is diagnosed from leaf biomass and specific leaf area index, which
varies as a function of leaf nitrogen concentration and plant functional type. To allow for leaf growth after dormancy there is
an imposed minimum leaf biomass. The model distinguishes 16 vegetation types (9 tree and 1 shrub types, 3 grass types and 3
crop types) alongside desert, rocks and permanent snow. Crops have the same phenology as grasses. A detailed description of
the terrestrial carbon cycle can be found in *Delire et al.* (2020).

### 2.3  Experimental setup

In this study, the S3 CRESCENDO simulations were used, characterized by transient $CO_2$, climate, and land-use forcing.
Each model spin-up is obtained by recycling climate mean and variability from the period 1901-1920, with the pre-industrial
(1860) atmospheric CO2 concentration until carbon pools and fluxes reach a steady state. The 1861-1900 period is simulated
using the same climate forcing as the spin-up, but with time-varying atmospheric $CO_2$ and land-use forcing. Finally, the LSMs
are forced over the 1901-2014 period with changing CO2, climate, and land-use forcing. All LSMs are commonly driven
by the atmospheric forcing reanalysis CRUNCEP version 7 (*Viovy*, 2018), and the land-use data is taken from the Land Use
Harmonization version 2 (*Hurtt et al.*, 2020). Note that the use of LUH2 land cover transitions differs across the models (see
model description). CRUNCEPv7 provides for 2m air temperature, precipitation, wind, surface pressure, shortwave radiation,
long-wave radiation, and air humidity.

Each LSM is run on different spatial resolutions (Table 1), but the outputs of these simulations are provided on a regular 0.5°
x 0.5° grid, over which simulations and observations are compared. The LAI monthly mean output from these simulations are
used in the present analysis.

### 2.4  Growing season analysis


The times of start and end of growing season (GSS and GSE, respectively) are evaluated using the Four Growing Season Types
(4GST) method introduced by *Peano et al.* (2019). 4GST has been shown to adequately capture the main global phenology
cycles for evaluation of LSMs.

The 4GST method allows to evaluate start and end of the growing season and the global spatial distribution of four main
growing season types: (1) evergreen (EVG); (2) single growing season peaking in summer (SGS-S); (3) single growing season
with summer dormancy (SGS-D); (4) two growing seasons (TGS). The EVG type is identified when relative changes in LAI
annual cycle are smaller than 25% of local LAI mean value. Note that GSS and GSE timings are not detected in EVG areas.
The other three types are identified based on LAI annual cycle shapes, LAI slopes and transition timings, and critical threshold,
as illustrated and summarized in Supplementary Figure 1. When one single growing season is identified, SGS-S and SGS-D are
discerned based on the peak-month (i.e. in the Northern Hemisphere (NH) SGS-S is detected when LAI peak occurs between
April and September, otherwise, SGS-D is detected, vice versa in the Southern Hemisphere (SH)). TGS, instead, is identified





when two growing seasons at least three-month-long are detected. Further details can be found in *Peano et al.* (2019). Note that in this analysis, the timings of the TGS GSS correspond to the GSS timings of the first growing season cycle, while the GSE are the second GSE timings, as described in *Peano et al.* (2019). The 4GST method is applied on monthly LAI data in this work,

instead of 15-day time-scale used in *Peano et al.* (2019). Given the monthly time window used in this analysis, difference up to one month can descend from the monthly time-frequency, which limits a more detailed bias assessment between LSMs and observations.

## 3 Results

### 3.1 Satellite data comparison

We inspect the main differences between LAI3g, MODIS and SENTINEL by plotting the spatial distribution of the four growing season types, GSS, and GSE (Figure 1).

The three products show a high consistency in the distribution of growing season types (agreement of about 80%, Table 2), with the main differences occurring in tropical regions, such as in Amazon and Congo basins, and in semi-arid areas, such as central Australia (Figures 1a,d,g). Compared to MODIS, LAI3g differs mainly in EVG regions (Table 2) due to an

underestimation of EVG areas in the Tropics (Supplementary Figure 2). These regions are characterized by high canopy density, which saturates to high LAI in the satellite data (e.g. *Myneni et al.*, 2002), resulting in limited seasonal variability. In addition, the AVHRR sensor used to derive LAI3g is less responsive to changes in vegetation compared to MODIS and SPOT/PROBA data (*Piao et al.*, 2020). Both LAI3g and SENTINEL differ from MODIS in areas featured by the TGS type (Table 2). The Horn of Africa is the only region where all three satellite products place a TGS type (Figure 1).

Larger differences among satellite products are found in the assessment of GSS and GSE (Figure 1), especially in the NH where LAI3g and SENTINEL clearly anticipate GSS (Figures 1e,h) with respect to MODIS. The three satellite products present a consistency similar to the one reached by the growing season type distribution (about 75%) when a one-month tolerance is considered (Table 3), since time-resolution of the products has been homogeneizied to one month (see Section 2.4).

Keeping these differences in mind, the MODIS data are in the following sections as the main observation reference given

its ability to capture seasonality of the different biomes (*Yan et al.*, 2016b). Comparisons with the other satellite products are presented in the supplementary material.

### 3.2 Growing season types distribution

The 4GST allows estimating the ability of each LSM in capturing the observed spatial distribution of the four growing season types (Figure 2).

In general, all the LSMs capture the single growing season that peaks in summer (SGS-S type) reasonably well, especially in the NH mid- and high-latitude regions. The majority of LSMs are also able to correctly represent the two growing seasons (TGS) in the Horn of Africa region (Figure 2). Most LSMs are unable to reproduce the observed growing-season-type distri-





bution in the SH, except for the evergreen (EVG) tropical areas. A partial exception is LPJ-GUESS, which shows large SGS-S type areas in South America, Southern Africa and Northern Australia, in agreement with MODIS.

LSMs used in this study are primarily able to capture the observed EVG and SGS-S regions with agreement between 36.0% and 95.4%, and between 44.3% and 79.5%, respectively (Table 2). In contrast, the TGS regions are seldom reproduced by LSMs, and the agreement rate with MODIS ranges between 0.4% and 19.1%, (Table 2). Overall, the CRESCENDO Multi-Model Ensemble (MME) mean reproduce the same MODIS growing season type distribution over about 69.5% of global land-surface area, with a 45.4% to 74.0% range among models (Figure 2 and Table 2). It is noteworthy that the evergreen

type is correctly detected in the broad-leaf evergreen tropical areas in both satellite observations and LSMs (Figure 2). On the contrary, the high-latitude needle-leaf evergreen regions are partially represented in LSMs, while satellite data do not catch these areas due to satellite limitations resulting from the impact of cloud and snow cover on light availability during the winter season (see Section 4.2).

     This initial evaluation highlights that LSMs have difficulties in accurately representing SH phenology. The correct location of

the less common types, i.e. single growing season with summer dormancy (SGS-D) and TGS, is as well hardly captured by the LSMs. Similar results are obtained when SENTINEL and LAI3g satellite observations are used as references (Supplementary Figures 3,4 and Supplementary Tables 1,2).

### 3.3   Variability of growing season start and end

4GST is then applied to evaluate the ability of LSMs to represent the GSS and GSE timing in vegetated areas not classified as

EVG-type (white regions in Figures 3 and 4 correspond to not-vegetated and EVG-type domains).

     On average at the global scale, LSMs approximately exhibit a disagreement of 0.6 months and 0.5 months in GSS and GSE, respectively, with LSMs simulating a later GSS and an earlier GSE, practically shortening the growing season by one month (Table 4). This bias is not evenly distributed around the globe. LSMs reproduce the correct growing season length in about 17% of the global land grid-cell, but sometimes growing season is affected by a shift in seasonality, as in the case of JULES-ES

(Table 3).

     Generally, the GSE simulated by the LSMs show a better agreement with MODIS (about 25% agreement in vegetated grid-cell, ranging from 4.9% to 26.4%, Table 3) compared to GSS timings (15.8% agreement in vegetated land grid-cell, ranging from 2.7% to 19.1%, Table 3). Considering a one-month-tolerance to account for the downgraded time-resolution, the agreement between LSMs and MODIS increases to ∼45% and ∼31%, respectively (Table 3).

LSMs exhibit larger agreement with MODIS GSS and GSE timings in the NH compare to the SH (Figures 3, 4 and Tables 3, 4). Only CLM 5.0 and LPJ-GUESS show similar results in both hemispheres (Table 3). In particular, LPJ-GUESS shows good skill (agreement with observation larger than 15%) in capturing both GSS and GSE timings in both hemispheres (Figures 2f, 3f, and 4f).

     LPJ-GUESS is the model that shows the highest agreement with MODIS (Table 3) and the lowest bias in average GSS and

GSE timings (0.4 and 0.1 months, respectively, Table 4). JULES-ES shows the lowest agreement with MODIS (Table 3) and the highest bias in the average GSS and GSE timings (1.2 and -2.3, respectively, Table 4). This result may be associated with





the representation of PFTs in the two models used to describe global vegetation. LPJ-GUESS, indeed, is the model featuring the largest number of PFTs, while JULES-ES uses the least (Table 1). Moreover, JULES-ES and LPJ-GUESS differ also on the details of the phenology parameterization. LPJ-GUESS features a more elaborate phenology description compared to JULES

(Sections 2.2.3, 2.2.5 and Table 1).

The two Community Land Model versions (i.e. CLM4.5, and CLM 5.0, Table 3) show very different outcomes, with CLM5.0 exhibiting larger biases in GSS and GSE timings compared to CLM4.5 (Figures 3b,c, and 4b,c, and Table 4). The two model versions differ in the crop representation, plant physiology, and phenology parameterization (Section 2.2). The implementation of an antecedent rain requirement trigger for stress deciduous PFTs (*Dahlin et al.*, 2015) help improved phenology in semi-arid

regions (e.g. the Sub-Sahara, Figures 3b,c, and 4b,c). Nonetheless, *Zhang et al.* (2019) show that the same upgrade influences the leaf senescence in temperate grasslands. On the other hand, the irrigation scheme in the CLM5.0 crop model allows for the improvement in crop-dominated regions, such as the Indian peninsula (Figures 3b,c, and 4b,c). Further differences occur between CLM 4.5 and CLM 5.0 (Figure 3b,c, and 4b,c), which could be ascribed to the changes in plant physiology, soil hydrology, and rooting profile. These features, indeed, impact on the representation of soil moisture, which has a significant

control on plant phenology (e.g. *Caldararu et al.*, 2012).

Similar results are obtained when LSMs are compared to LAI3g and SENTINEL satellite observations (Supplementary Figures 6-9, and Supplementary Tables 3-6).

### 3.4 Latitudinal variability

The MME zonal average shown in Figure 5 highlights the LSMs' abilities and limitations in simulating the observed GSS and

GSE timings at different latitudes.

The GSS bias ranges between -1.8 months (earlier GSS) just south of the Equator and +2.0 months (delayed GSS) south of 50°S (Figure 5a). The GSE bias ranges between -3.0 months in the 0-10°N latitudinal band and +1.3 months in the southern sub-tropics. The CRESCENDO LSMs correctly simulate the GSE timings north of 60 °N. The Spearman correlation of the GSS and GSE latitudinal distributions is 0.67±0.07 and 0.51±0.11, respectively. These values are significant at the 95% level

based on a Monte Carlo approach.

In the NH mid- and high-latitude, the LSMs' GSS exhibit an average delay of up to 1.6 months, especially in North America (Supplementary Figure 10a). Note that differences among satellite data occur in the NH mid- and high-latitude, highlighting potential differences among these three products (see Section 4.2). Large LSM biases in NH tropical region GSE timings and southern sub-tropical GSS timings are driven by premature values in Africa (Supplementary Figures 10c,d). These discrep-

ancies may derive from difficulties in the LSM's ability to simulate the observed phenology type in Africa (Figure 2, and Supplementary Figure 11).

Observed latitudinal distributions highlight an increasing northward trend in the NH mid-latitude GSE timings (GSE around May-June at ∼20°N and around September-October at ∼40°N, Figure 5b), and an increasing southward trend in the 30-55°S latitudinal band (GSS around July at ∼30°S and around September at ∼55°S, Figure 5a). Similar trends are reproduced by

the LSMs, but with a higher magnitude (Figure 5). In the NH, the difference between simulated and observed trends may be





driven by an overestimated influence of radiation and temperature on leaf senescence in LSMs. In the SH, the discrepancies between observed and modeled trends may be related to relatively large phenology variability in the SH associated with the small vegetated land area in this hemisphere.

### 3.5 Regional variability

To assess sources of biases in the LSMs, different biomes derived from the ESA CCI land cover map (*Li et al.*, 2018, Figure 6a) are investigated.

The GSS timings are generally delayed compared to observations, except for the Broadleaf Deciduous Tree (BDT) and the Broadleaf Deciduous Shrub (BDS) biomes. In BDT-dominated regions, the Multi-Model Ensemble mean (MME) falls within the observational range (Figure 6c), while BDS areas show slightly earlier GSS timings on average (Figure 6f). Although the 390 overall skill of the MME is quite reasonable for the BDS biome, a large spread among LSMs exists: three LSMs show later GSS (CLM4.5, LPJ-GUESS, and ORCHIDEE), three LSMs display earlier GSS (JSBACH, JULES, and ISBA-CTRIP), and one LSMs falls within the observational range (CLM5.0, Figure 6f). The BDS-dominated regions are semi-arid and transition areas, where LSMs' parameterization could be more sensitive to climate conditions and parameter selection.

GSE timings display heterogeneous outcomes (Figure 7). On average, BDS and BDT (Figures 7f,c) display delayed GSE 395 values, while Broadleaf Evergreen Tree and Crop (BET and Crop, Figures 7b,h) show earlier GSE timings compared to observations. The Needleleaf Trees both evergreen and deciduous (NET and NDT, respectively) and Grass exhibit values within the observed range (Figures 7d, e, g). Similar to GSS timings, a high spread among LSMs exists for the BDS biome (Figure 7f). Large disagreement among LSMs also occurs for Crop-dominated areas (Figure 7h). This result highlights the need for further investigation on the representation of crop phenology in the LSMs.

In general, LSMs show a higher agreement in representing GSS timings compared to GSE timings. Consequently, the different approaches used to describe the start of the growing season are relatively consistent among LSMs. In comparison, the representation of the end of vegetative season requires further investigation and development.

## 4 Discussion

### 4.1 Land surface models

The plant phenology growing season start and end are mainly triggered by changes in solar radiation, temperature, and soil moisture conditions (e.g. *Caldararu et al.*, 2012; *Zeng et al.*, 2013; *Tang and Dubayah*, 2017). State-of-the-art LSMs represent the phenological transitions using different parameterizations based on the climate conditions (Section 2.2). Many of these parameterizations (see Section 2.2) are based on values derived from localized observations (e.g. *White et al.*, 1997; *Thornton et al.*, 2002; *Savoy and Mackay*, 2015). Consequently, the phenology parameters are calibrated on specific regions of the globe, 410 which may be one reason for the large spread of values seen in the present analysis.





Generally, phenology calibration areas are located in the NH, where LSMs exhibit better results and larger coherence compared to the SH. Among the LSMs evaluated here, LPJ-GUESS, CLM4.5, and ORCHIDEE show good skill (agreement with observation larger than 15%) in the SH (Table 3). On the other hand, CLM5.0 and JULES-ES do not reach such agreement in the NH (Table 3). High skill (agreement with observation larger than 20% for at least one timing) in the NH are obtained by
CLM4.5, ORCHIDEE, and ISBA-CTRIP (Table 3). The different performance between models can occur from differences in phenology parameterization as well as different vegetation cover types (Plant and Crop Functional types), soil characterization, and initial spatial resolution (Table 1).

Among the LSMs evaluated here, JULES-ES shows relatively lower skill in simulating GSS and GSE timings compared to the other LSMs (Table 3). This result may be ascribed to the smaller number of PFTs (see Table 1) and details of the phenology
parameterization that characterize this LSM (Section 2.2.3 and Table 1). JSBACH accounts for a similar number of PFTs (Table 1), but features a more complex phenology scheme (Section 2.2.4 and Table 1), and its skill in reproducing GSS and GSE timings is higher than that of JULES-ES (Table 3). Similar to JSBACH, ORCHIDEE feature a PFT-oriented phenology scheme (Section 2.2.6 and Table 1), which contributes to the high skill noted for ORCHIDEE.

CLM 4.5, CLM 5.0, and LPJ-GUESS use three phenology schemes: (1) evergreen; (2) seasonal-deciduous; (3) stress-
deciduous (Sections 2.2.1, 2.2.2, 2.2.5). Among these schemes, the seasonal-deciduous one employs calendar thresholds (summer and winter solstices and day-length threshold in CLM, and fixed 210-day phenology in LPJ-GUESS) that may improve the results of LPJ-GUESS and CLM 4.5. On the other hand, this may mean that the seasonal-deciduous type may be less responsive to future climate change.

Contrary to the other LSMs, ISBA-CTRIP uses the daily leaf carbon balance to simulate plant phenology, and it reaches
good skill (Tables 3, 4). Consequently, ISBA-CTRIP highlights the opportunity to attain results aligned with the other LSMs using leaf carbon availability instead of climatic conditions.

The improvement of the phenology parameterization can lead to better representation of vegetation in the LSMs. However, other vegetation features affect the plant phenology representation, as in the case of the two CLM versions. CLM4.5 and CLM5.0 share similar phenology parameterization (Sections 2.2.1 and 2.2.2) but differ in the crop irrigation scheme, soil and
plant hydrology, and carbon and nitrogen cycling (*Lawrence et al.*, 2018). Since soil moisture has a significant control on plant phenology (e.g. *Caldararu et al.*, 2012), CLM5.0 revision of stomatal response to rising $CO_2$ concentrations through a new Medlyn stomatal conductance scheme (*Fisher et al.*, 2019; *Medlyn et al.*, 2011) and the use of a revised mechanistically based soil evaporation parameterization that accounts for the rate of diffusion of water vapor through a dry surface layer (*Swenson and Lawrence*, 2014) are likely to be principal sources of differences between CLM5.0 and CLM4.5.
In general, this comparison highlights the complexity of vegetation phenology modelling and the strong inter-linkages between climate, hydrology, soil, and plants.

## 4.2 Satellite data

Satellite-based LAI datasets have been used in this work as a benchmark for the evaluation of the LSMs' phenology performance globally. However, satellite observations present some caveats and uncertainties (e.g. *Myneni et al.*, 2002; *Fang et al.*,





2013; *Jiang et al.*, 2017). For this reason, three separate satellite LAI products obtained from different acquisition sensors (namely AVHRR for LAI3g, MODIS for MODIS LAI, and SPOT/PROBA VEGETATION for SENTINEL) have been used in this study. The comparison between these datasets shows major issues associated with LAI products derived from satellite reflectance observations. For example, large differences between LAI3g, MODIS, and SENTINEL occur at high latitudes and in tropical regions (Figure 5), where thick clouds and snow cover can affect the data reconstruction (e.g. *Delbart et al.*,

2006; *Kandasamy et al.*, 2013; *Yan et al.*, 2016b). LAI satellite data are also affected by the applied gap-filling algorithms, which could create spurious seasonal cycles as well as smooth the observed phenology season (e.g. *Kandasamy et al.*, 2013; *Chen et al.*, 2017). In addition, the observed reflectance saturates in regions characterized by dense canopies, reaching the LAI upper limits (e.g. *Myneni et al.*, 2002; *Maignan et al.*, 2011) of approximately 7.0 $m^2/m^2$ in MODIS and LAI3g (*Myneni et al.*, 2002). This issue can affect the identification of growing season cycles in thickly forested areas leading to possible

overestimation of evergreen type detection.

### 4.3 4GST limitations

Contrary to previous phenology analysis that focused on specific biomes (e.g. *Dahlin et al.*, 2015) or NH mid- and high-latitude regions (above 30° N *Anav et al.*, 2013; *Murray-Tortarolo et al.*, 2013), 4GST accounts explicitly for different phenology types on the global scale. In particular, it takes into account SGS-D and TGS types that were neglected in previous analyses (*Murray-

Tortarolo et al.*, 2013) due to their reduced coverage (Table 2).

Regions with multiple growing seasons per year (TGS) are difficult to capture on a global scale, despite their important influence on climate (e.g. *Zhang et al.*, 2003; *Dalmonech and Zaehle*, 2013; *Peano et al.*, 2019). The state-of-the-art LSMs, indeed, exhibit a low skill in reproducing this specific growing season type (Figure 2 and Table 2). Two growing seasons usually occur in regions characterized by two separate rain seasons, in semi-arid areas or in cropland regions (e.g. *Zhang et al.*,

2003, 2005; *Martiny et al.*, 2006). In this analysis, observations and LSMs, except for JULES-ES and ORCHIDEE, agree on a TGS type only in the Horn of Africa. This region features two distinct precipitation seasons (e.g. *Liebmann et al.*, 2012; *Peano et al.*, 2019), which trigger the TGS phenology type.

Cropland areas can present multi growing season behavior because of irrigation and crop rotation, such as in South Asia, and China (e.g. *Wu et al.*, 2010; *Gumma et al.*, 2016). Unlike the Horn of Africa, these regions are not captured as TGS in

the present analysis due to assumptions and limitations within LSMs, for example CLM4.5 represents all annual crops by a generic C3 PFT (Section 2.2.1, and Table 1), and 4GST assumptions. 4GST TGS type detection adopts a minimum length of three months to detect a growing season. This assumption derives from the need to avoid the detection of small oscillations within the same growing seasons (*Peano et al.*, 2019). Consequently, this assumption affects the model recognition of multiple growing seasons, especially in cropland areas. South Asia, for instance, is characterized by different timing and phenology

intensity for each crop growing season (*Gumma et al.*, 2016). Therefore, only some specific crops can be detected based on the 4GST assumptions and growing season signature (*Gumma et al.*, 2016). *Wu et al.* (2010) distinguish multiple growing seasons in China using local maximum and detection threshold. This method improves the multi growing season identification, especially for the crops identified by a strong phenology cycle but may exclude crop characterized by a weak phenological





cycle. For these reasons, more specific analyses of semi-arid and crop regions based on higher spatial and temporal data are
needed and will be the focus of a future study.

Besides, LSMs' crop phenology parameterizations require further development to improve the description of each specific crop.

LSMs evaluate phenology at PFT level, but the final LAI values are returned at the grid-cell level. For this reason, a more detailed evaluation of the parameterization would require using PFT-level values. However, global coverage of observed PFT level
phenology values are missing, making this analysis limited to specific biomes, such as through PhenoCAM data (*Richardson et al.*, 2018). This analysis will be the focus of a future work.

## 5   Conclusions

This study evaluates the ability of seven European Land Surface Models (LSMs) to reproduce the timings of start and end of the plant growing season at the global level. The assessment is performed based on the novel four growing season types
methodology, and uses a set of three satellite observation products as a benchmark to account for some of the uncertainty in observations.

In general, LSMs exhibit better agreement with observations in the NH compared to the SH, where large variability associated with the small vegetated land area is present. LSMs also show higher ability to simulate the timing of growing season end compared to the timing of the growing season start. On average, LSMs show 0.6 months delay in estimating start of the
growing season and about 0.5 months premature end of the growing season, leading to about one month shorter phenology active season. High discrepancies between LSMs and satellite products are noted for growing season start (GSS) timings in the region poleward of 50°S, where simulated GSS is delayed by about two months. The growing season end (GSE) shows high differences between LSMs and observations in the 0-10°N latitudinal band, where LSMs simulate a three-month earlier GSE. On the contrary, the LSMs accurately simulate the GSE timings poleward of 60 °N and the GSS in the 30-40°S and 10-30°N
latitudinal bands. At the biome scale, LSMs correctly simulate the GSS timings in Broad-leaf Deciduous Trees dominated areas and the GSE timings in Grass, Needle-leaf Deciduous Trees, and Needle-leaf Evergreen Trees regions. High uncertainty remains in the Broad-leaf Deciduous Shrubs and Crop dominated areas.

Despite a lower ability of LSMs to represent SH phenology, LPJ-GUESS, CLM4.5, and ORCHIDEE show reasonably good outcomes in these regions. In the NH, high skill is achieved by CLM4.5, ORCHIDEE, and ISBA-CTRIP. Uncertainties and
spread among LSMs remain, which might affect our understanding of present-day and future impact of land and vegetation interactions with the climate and carbon cycle. Therefore, further improvements in LSMs will be necessary.

Improvements in the phenology parameterization can lead to better representation of vegetation in the LSMs. However, phenology in LSMs is influenced by vegetation and hydrological parameterizations and land surface boundary conditions (e.g. PFT distribution), as shown by the CLM4.5 and CLM5.0 phenology differences.
This study highlights the complexity of vegetation phenology modeling and the strong inter-linkages between climate, hydrology, soil, and plants, which need further details and generalization inside the LSMs code.





*Code and data availability.* The LAI3g satellite observation data are available from R. Myneni (http://sites.bu.edu/ cliveg/datacodes/); the MODIS satellite observation data are available from T. Park; the SENTINEL satellite observation data are available from COPERNICUS (https://land.copernicus.eu/global/products/lai); the atmospheric forcing, CRUNCEP v7, are available from N. Viovy (https://rda.ucar.edu/

datasets/ds314.3/); the land surface models simulations are part of CRESCENDO project and they are stored at the CEDA JASMIN service (http://www. ceda.ac.uk/); the 4GST python script is available online (https://github.com/ CMCC- Foundation/4GST.git)

*Author contributions.* DP wrote the paper, performed the analysis, and provided CLM 4.5 data; DH and SM were strongly involved in the discussion of result and in drafting the manuscript; TP provided MODIS data; DW provided LPJ-GUESS data; YF and HL provided CLM 5.0 data; AW provided JULES-ES data; EJ and CD provided ISBA-CTRIP data; PP provided ORCHIDEE data; JEMSN provided JSBACH

data; all co-authors discussed the results and contributed to writing the manuscript. Authors after SM are listed in alphabetic order.

*Competing interests.* The authors declare no competing interests.

*Acknowledgements.* This work has received funding from the European Union's Horizon 2020 research and innovation program under Grant Agreement 641816 (CRESCENDO). DH also received support from the Met Office Hadley Centre Climate Programme (HCCP) funded by BEIS and Defra. DW acknowledges financial support from the Strategic Research Area MERGE (Modeling the Regional and Global Earth

System - www.merge.lu.se), and from the Swedish national strategic e-science research program eSSENCE (http://essenceofescience.se/) LPJ-GUESS simulations were performed on the Tetralith supercomputer of the Swedish National Infrastructure for Computing (SNIC) at Linköping University, under project no. SNIC 2018/2-11 (S-CMIP). TP acknowledges support from NASA's Carbon Monitoring System program (80NSSC18K0173-CMS).



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





**Table 1.** Grid spatial resolution used for each land surface model (LSM) and brief summary of their main features. PFT stands for Plant Functional Type and CFT stands for Crop Functional Type.

| LSM | Original Resolution | PFT number | Soil Level | Crop (CFT number) | Phenology scheme |
|---|---|---|---|---|---|
| CLM 4.5 | 1.25° x 0.9375° | 15 | 15 | 1 C3 | evergreen; seasonal-deciduous; stress-deciduous |
| CLM 5.0 | 0.5° x 0.5° | 16 | 20 | 2 C3 | evergreen; seasonal-deciduous; stress-deciduous |
| JULES-ES | 1.875° x 1.25° | 13 | 4 | 1 C3 and 1 C4 | Deciduous broadleaf and needleleaf |
| JSBACH | 1.9° x 1.9° | 12 | 5 | 1 C3 and 1 C4 | evergreen; summergreen; raingreen; grasses; tropical and extra-tropical crops |
| LPJ-GUESS | 0.5° x 0.5° | 25 | 2 | 3 C3 and 2 C4 | evergreen; seasonal-deciduous; stress-deciduous |
| ORCHIDEE | 0.5° x 0.5° | 15 | 11 | 1 C3 and 1 C4 | deciduous; dry and semi-arid; grasses and crops |
| ISBA-CTRIP | 1° x 1° | 16 | 14 | 1 C3 and 1 C4 | leaf biomass |

**Figure 1.** Global climatological (averaged over 2000-2011) distribution of (a) the four main growing season modes, (b) growing season start (GSS) timings, and (c) growing season end (GSE) timings for MODIS version 6. The other panels show the comparison between MODIS and LAI3g (second row), and SENTINEL (third row). In particular, panels (d) and (g) show the areas characterized by the same phenology types in both MODIS and LAI3g and SENTINEL, respectively; panels (e) and (h) exhibit the difference in GSS timings while panels (f) and (i) display the differences in GSE timings. In panels (d) and (g) white areas represent non-vegetated areas and regions of disagreement between MODIS and LAI3g and SENTINEL. White areas in panels (b),(c),(e),(f),(h), and (i) show evergreen and non-vegetated areas.





**Table 2.** The fraction of land grid-cell in agreement with MODIS for each satellite product and each land surface model in the four growing season types. Values are reported in percentage, and refer to the colored regions in Figures 1d,g and 2.

|       | LAI3g | SENTINEL | CLM 4.5 | CLM 5.0 | JULES-ES | JSBACH | LPJ-GUESS | ORCHIDEE | ISBA-CTRIP | MME |
|-------|-------|----------|---------|---------|----------|--------|-----------|----------|------------|------|
| EVG   | 10.8  | 78.3     | 58.1    | 72.9    | 95.4     | 36.0   | 55.2      | 72.3     | 75.3       | 51.0 |
| SGS-S | 84.4  | 89.9     | 64.3    | 44.3    | 47.9     | 71.5   | 73.8      | 79.5     | 74.1       | 77.2 |
| SGS-D | 68.3  | 80.9     | 47.6    | 36.3    | 7.1      | 50.6   | 33.2      | 33.1     | 58.8       | 23.8 |
| TGS   | 37.2  | 35.9     | 19.1    | 16.9    | 0.4      | 13.9   | 15.4      | 7.9      | 12.9       | 0.7  |
| Total | 75.4  | 86.4     | 61.2    | 45.4    | 48.3     | 65.2   | 68.1      | 74.0     | 71.1       | 69.5 |



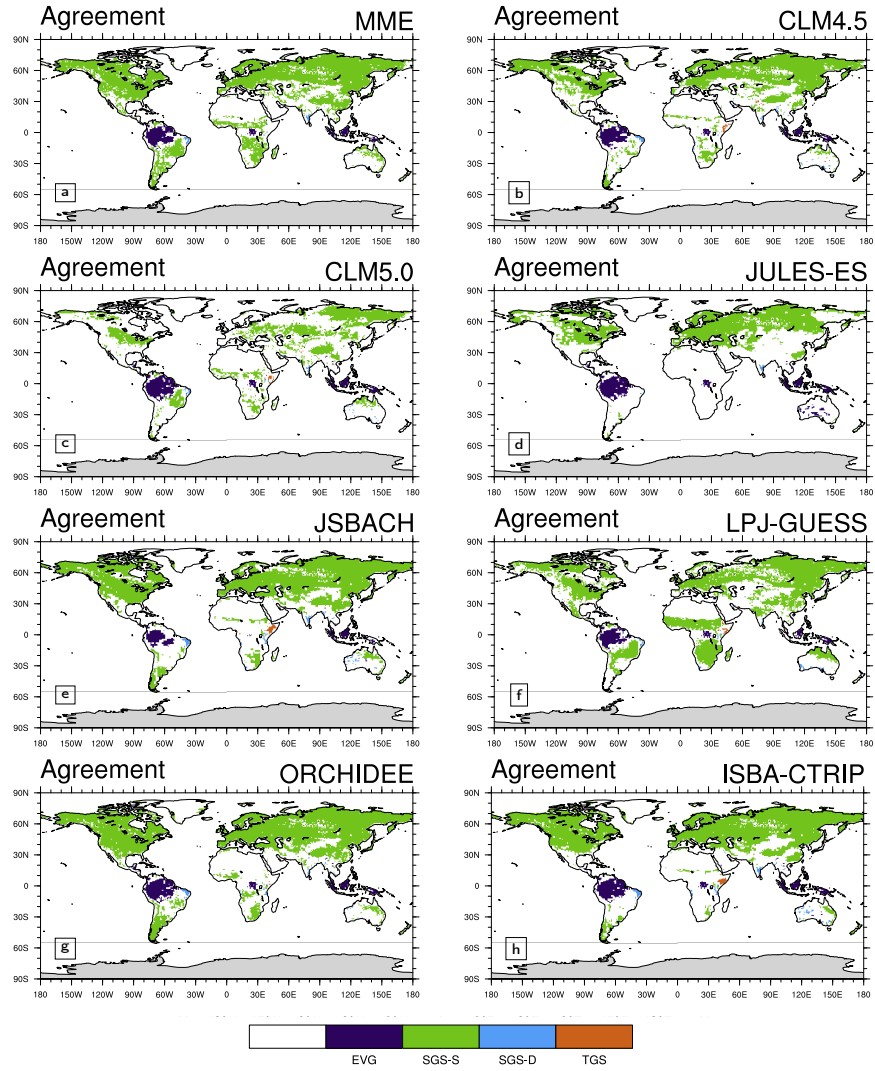

**Figure 2.** Global climatological (averaged over 2000-2011) distribution of the four main growing season modes for (a) MME; (b) CLM 4.5; (c) CLM 5.0; (d) JULES-ES; (e) JSBACH; (f) LPJ-GUESS; (g) ORCHIDEE; (h) ISBA-CTRIP. Only the areas characterized by the same type of MODIS (Figure 1a) are shown. These common areas are called agreement regions. Index values: (1, purple) evergreen; (2, green) single season with summer LAI peak; (3, cyan) single growing season with summer dormancy; (4, orange) two growing seasons type. White regions are for disagreement areas.





**Table 3.** The fraction of land grid-cell in agreement with MODIS for LAI3g, SENTINEL, each land surface model, and multi-model ensemble mean (MME) in growing season start (GSS) and growing season end (GSE) timings. Values are reported in percentages for global, North Hemisphere (NH), and South Hemisphere (SH). Green shaded areas in Figures 3 and 4. The last row reports the fraction of land grid-cell in agreement with MODIS for each land surface model in growing season length. The values in brackets give the percentage of global, NH, and SH with a maximum difference of one month.

| | LAI3g | SENTINEL | CLM 4.5 | CLM 5.0 | JULES-ES | JSBACH | LPJ-GUESS | ORCHIDEE | ISBA-CTRIP | MME |
|---|---|---|---|---|---|---|---|---|---|---|
| GSS | 37.9 | 43.7 | 14.6 | 6.5 | 2.7 | 16.7 | 17.3 | 19.1 | 16.0 | 15.8 |
| | (74.0) | (74.2) | (36.1) | (21.6) | (15.0) | (41.3) | (45.1) | (44.2) | (43.1) | (31.3) |
| GSE | 45.1 | 34.0 | 23.5 | 6.5 | 4.9 | 12.6 | 20.6 | 26.4 | 19.9 | 25.1 |
| | (80.1) | (70.5) | (44.8) | (30.0) | (14.7) | (38.5) | (60.5) | (63.8) | (62.5) | (44.6) |
| GSS NH | 39.2 | 39.7 | 15.1 | 5.7 | 3.2 | 17.1 | 17.6 | 18.9 | 16.5 | 16.3 |
| | (76.4) | (72.6) | (37.8) | (20.6) | (17.7) | (43.4) | (44.5) | (43.9) | (44.5) | (32.9) |
| GSE NH | 47.3 | 29.6 | 25.4 | 6.5 | 6.0 | 14.4 | 19.9 | 29.9 | 21.7 | 27.4 |
| | (83.6) | (68.1) | (46.9) | (31.2) | (17.7) | (43.5) | (61.0) | (69.2) | (68.1) | (47.5) |
| GSS SH | 32.6 | 60.7 | 12.6 | 10.0 | 0.7 | 14.8 | 16.0 | 19.8 | 14.1 | 13.3 |
| | (63.8) | (81.0) | (28.8) | (25.9) | (3.4) | (32.4) | (47.7) | (45.6) | (36.8) | (24.8) |
| GSE SH | 35.9 | 52.7 | 15.0 | 6.6 | 0.2 | 5.1 | 23.4 | 11.6 | 12.0 | 15.5 |
| | (64.8) | (80.9) | (35.7) | (24.9) | (1.8) | (17.5) | (58.2) | (40.4) | (38.7) | (32.1) |
| Length | 26.7 | 28.9 | 7.9 | 10.3 | 9.5 | 17.4 | 9.2 | 12.7 | 13.7 | 16.7 |
| | (60.6) | (54.2) | (28.4) | (33.7) | (27.0) | (45.3) | (31.9) | (39.1) | (35.2) | (35.2) |





**Table 4.** Average difference between MODIS and LAI3g, SENTINEL, each land surface model, and multi-model ensemble mean (MME) in Growing Season Start (GSS) and Growing Season End (GSE) timings. Values are reported in months for global, North Hemisphere (NH), and South Hemisphere (SH). Positive values stand for later timings and negative values correspond to earlier timings.

| | LAI3g | SENTINEL | CLM 4.5 | CLM 5.0 | JULES-ES | JSBACH | LPJ-GUESS | ORCHIDEE | ISBA-CTRIP | MME |
|---|---|---|---|---|---|---|---|---|---|---|
| GSS | 0.25 | 0.30 | 0.54 | 0.81 | 1.23 | 0.35 | 0.37 | 0.64 | 0.44 | 0.56 |
| GSE | -0.16 | -0.31 | -0.30 | -1.15 | -2.26 | -0.28 | 0.14 | -0.10 | -0.30 | -0.49 |
| GSS NH | 0.41 | 0.38 | 0.95 | 1.47 | 1.42 | 0.95 | 0.44 | 0.69 | 0.97 | 0.98 |
| GSE NH | -0.31 | -0.48 | -0.47 | -1.69 | -2.58 | -0.69 | 0.10 | -0.29 | -0.59 | -0.84 |
| GSS SH | -0.42 | -0.01 | -1.31 | -2.18 | -1.80 | -2.48 | 0.11 | 0.41 | -2.04 | -1.23 |
| GSE SH | 0.46 | 0.39 | 0.42 | 1.34 | 2.94 | 1.66 | 0.34 | 0.82 | 1.03 | 1.00 |



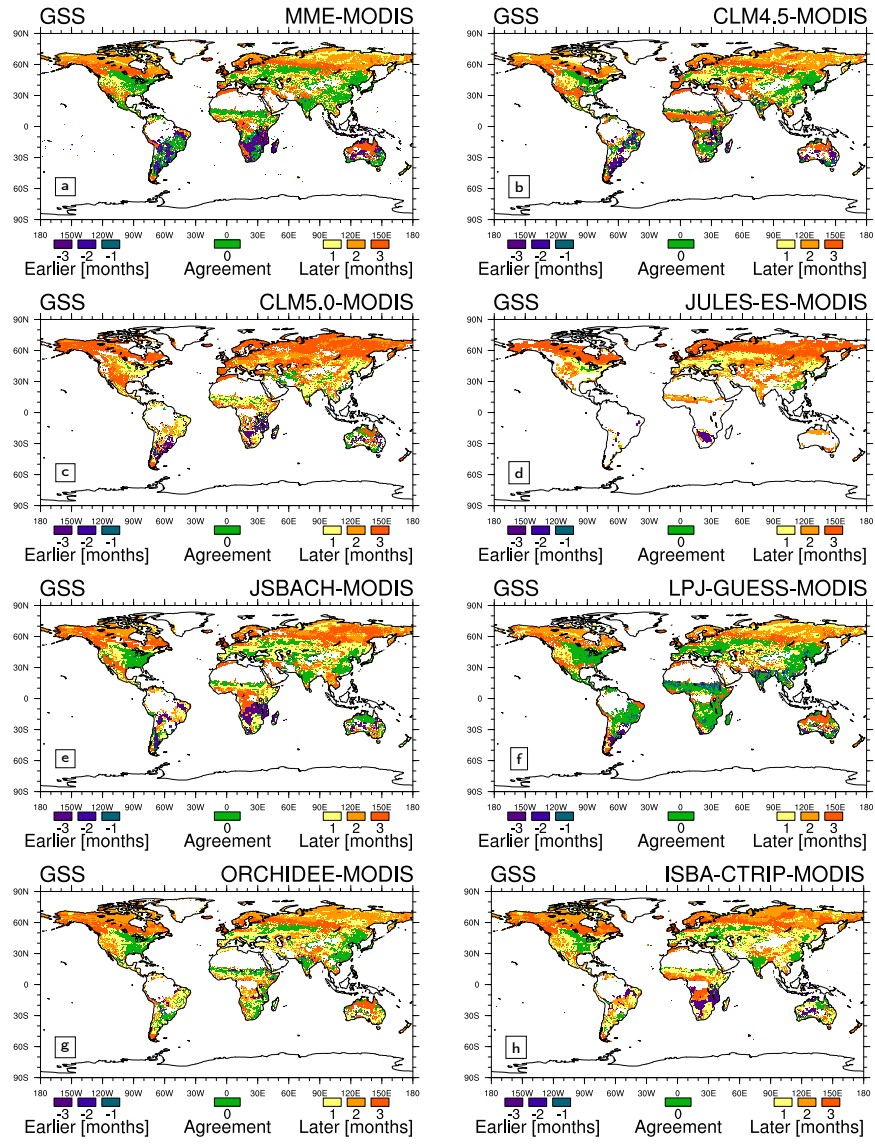

**Figure 3.** Global climatological (averaged over 2000-2011) differences in growing season start timings (GSS) between (a) Multi-Model Ensemble mean (MME); (b) CLM 4.5; (c) CLM 5.0; (d) JULES-ES; (e) JSBACH; (f) LPJ-GUESS; (g) ORCHIDEE; (h) ISBA-CTRIP and MODIS (Figure 1b). The green regions represent areas of agreement between MODIS and LSMs. Yellow-red colors correspond to areas where models timings are later compared to MODIS, while blue-violet colors correspond to areas where models timings are earlier compared to MODIS. Regions where GSS timings are not computed, such as non-vegetated and evergreen areas, are in white. Note that the GSS in the TGS regions corresponds to the GSS of the first growing season cycle.

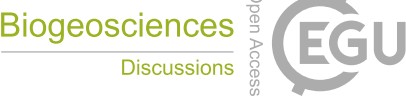



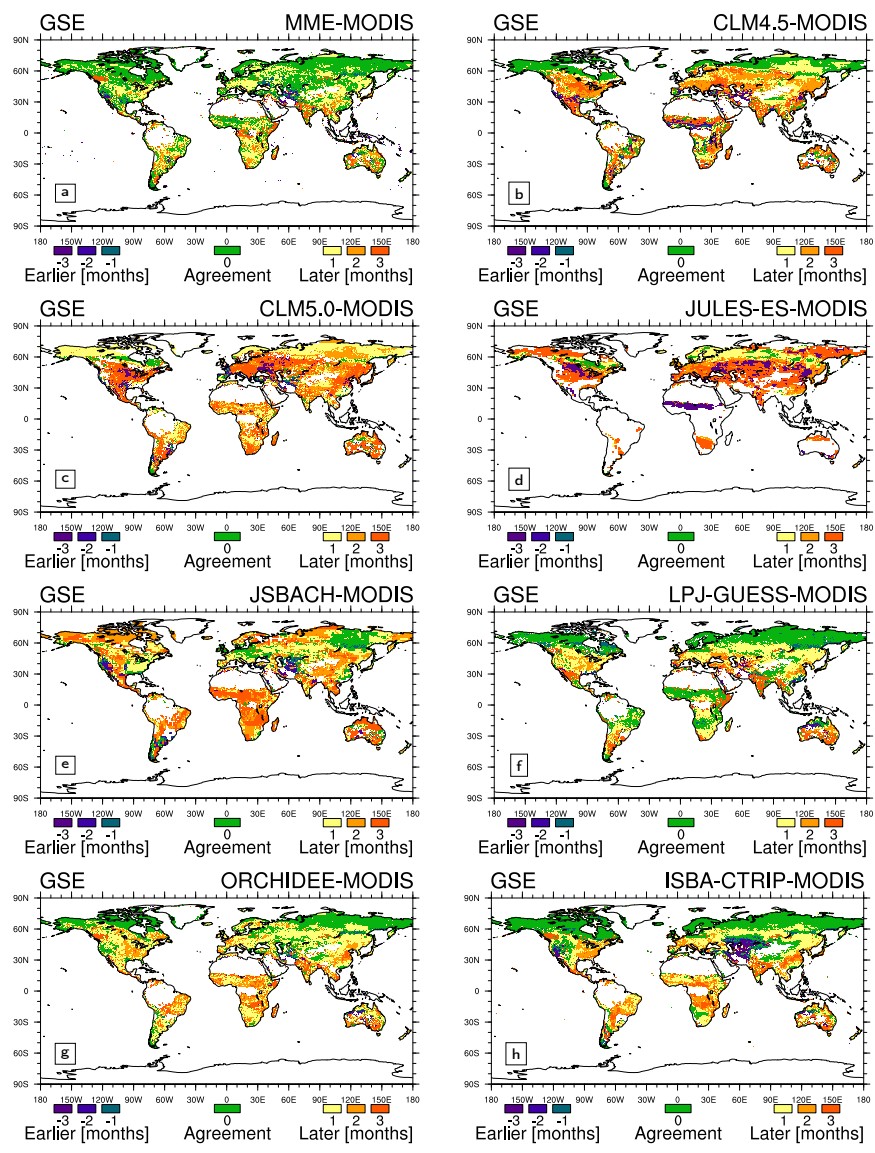

**Figure 4.** As Figure 3 but for growing season end (GSE) timings between MODIS (Figure 1c) and Multi-Model Ensemble mean (MME) and LSMs. Note that the GSE in the TGS regions corresponds to the GSE of the second growing season cycle.



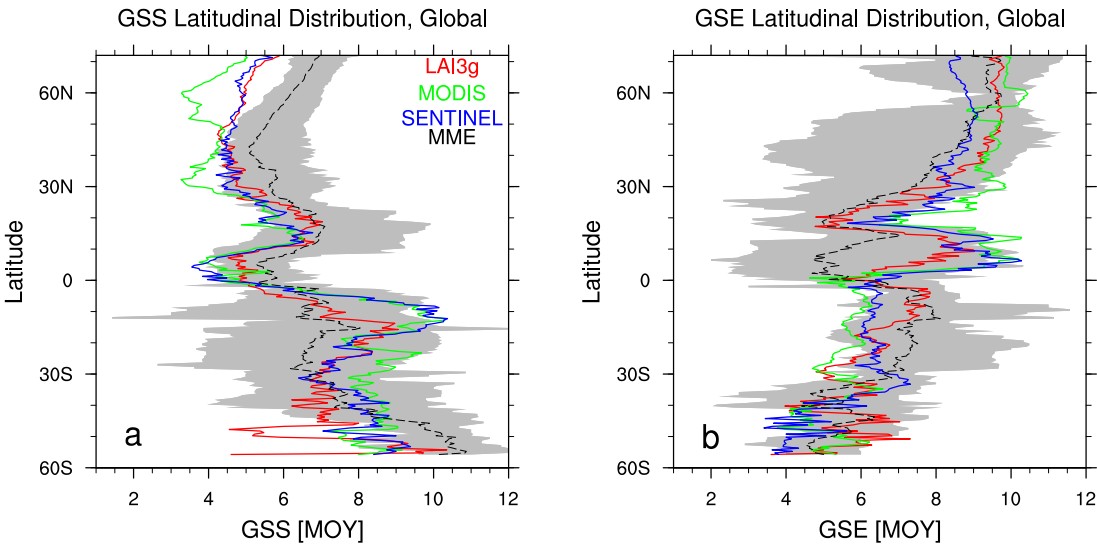

**Figure 5.** Zonal mean (a) growing season start (GSS) and (b) growing season end (GSE) timings for LAI3g (red lines), MODIS (green lines), SENTINEL (blue lines), and multi-model ensemble mean (black dashed line). The grey regions show the multi-model ensemble spread. Values are reported as month of the year (MOY), and the latitudinal coverage goes from 56°S to 75°N, which is the range covered by SENTINEL.





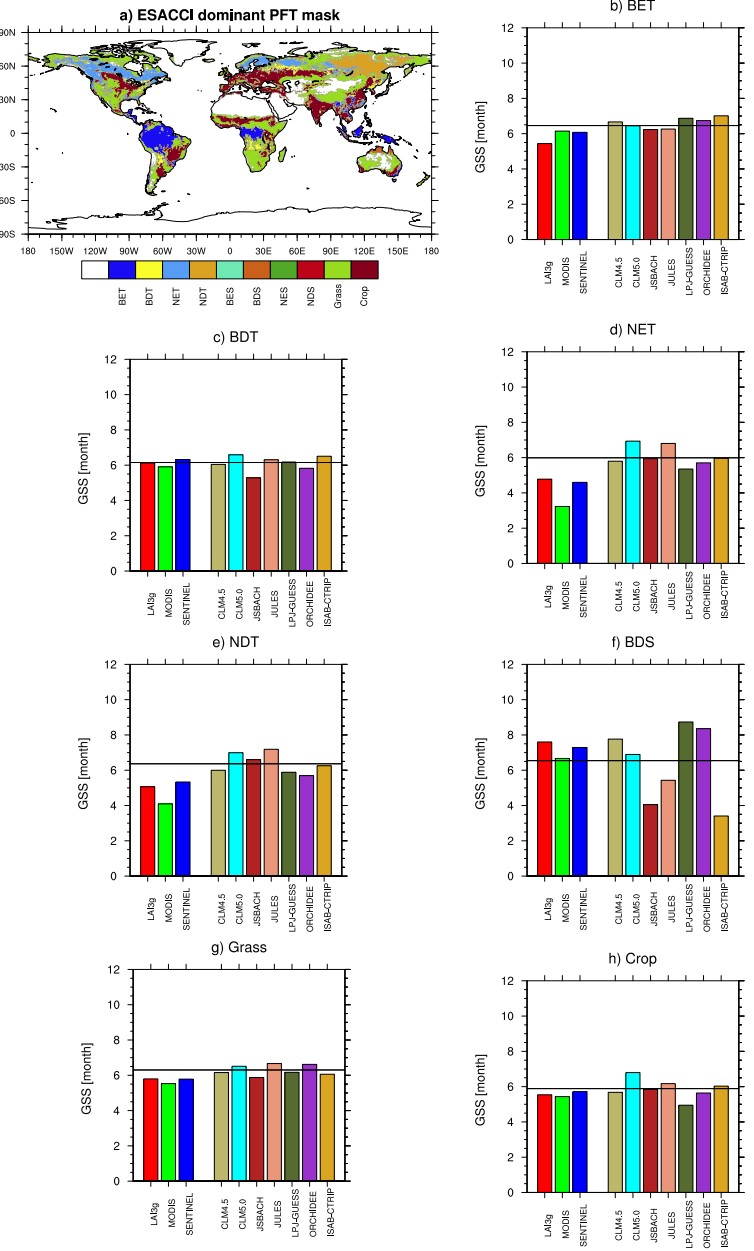

**Figure 6.** (a) Global distribution of the main land cover types for the 2000-2011 period based on ESA-CCI data (*Li et al.*, 2018). Comparison in growing season start (GSS) timings between satellite products (LAI3g, red columns; MODIS, green columns; SENTINEL, blue columns) and land surface models (LSMs: CLM4.5, dust columns; CLM5.0, cyan columns; JSBACH, dark red columns; JULES, pink columns; LPJ-GUESS, dark green columns; ORCHIDEE, purple columns; ISBA-CTRIP, dark yellow columns) in (b) Broad-leaf Evergreen Trees (BET); (c) Broad-leaf Deciduous Trees (BDT); (d) Needle-leaf Evergreen Trees (NET); (e) Needle-leaf Deciduous Trees (NDT); (f) Broad-leaf Deciduous Shrubs (BDS); (g) grass-covered areas (Grass); (h) crop-covered areas (Crop). Note that no area is dominated by Broad-leaf Evergreen Shrubs (BES), Needle-leaf Evergreen Shrubs (NES), or Needle-leaf Deciduous Shrubs (NDS) biome. The black horizontal lines in panels from (b) to (h) display the multi-model ensemble mean values.



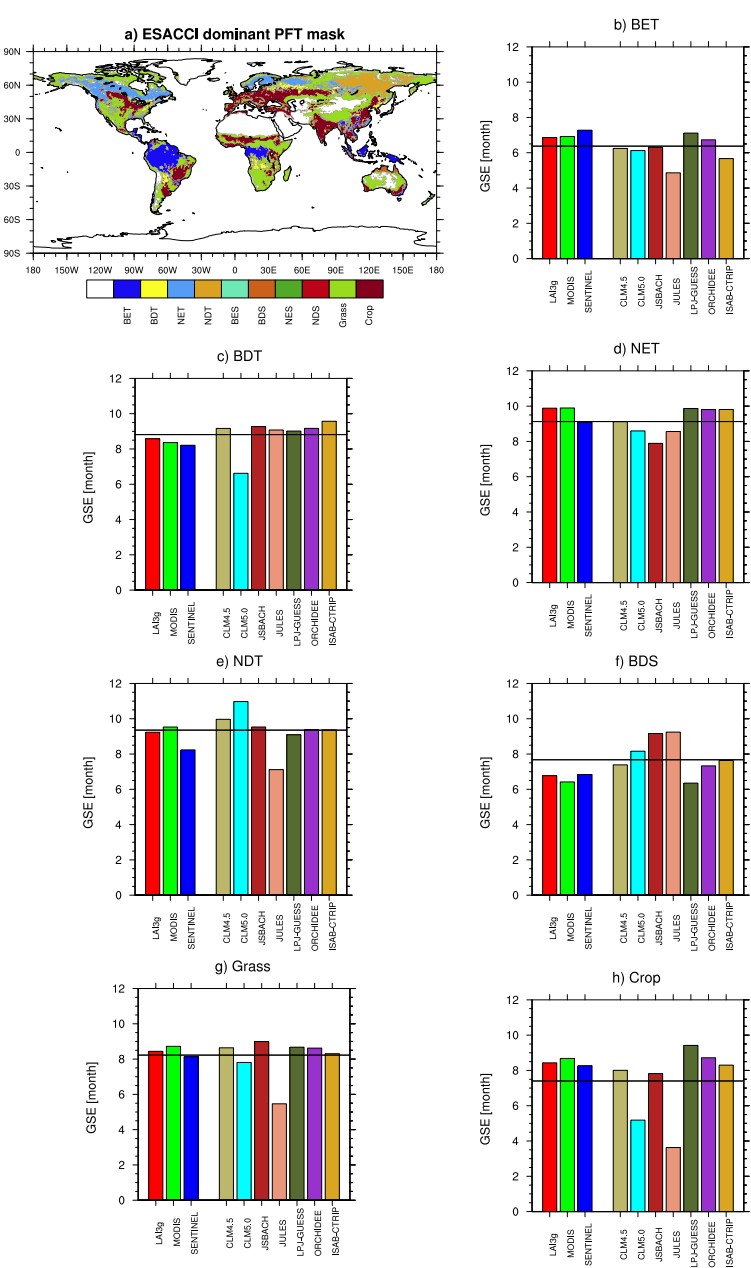

**Figure 7.** As Figure 6 but for growing season end (GSE) timings.