# Peer review of "Plant phenology evaluation of CRESCENDO land surface models."

_Biogeosciences, 2020_

## Referee Comment (RC1) · Anonymous Referee #1 · 23 Sep 2020

The study evaluated the plant phenology simulated by CRESCENDO land surface models using satellite observational LAI products. Specially, the 4GST method was applied to extract the times of start and end of growing season based on the simulated and remote sensing monthly LAI values. Then, the growing season types, variability of growing season start and end, latitudinal variability, and reginal variability were compared between the model simulations and satellite observations. Recommendations were also given for future model improvements. In general, the manuscript was written well, organized well, and the results were summarized clearly and interesting. So, I think the manuscript can be accepted for publication on the journal. Only one main remark is that the description of the phenology schemes of the models. As we know,

the phenology schemes in the models are quite different, in terms of their parameterizations of solar radiation, day-length, temperature, and soil moisture conditions. In section 2.2, the description of phenology schemes makes me a little bit hard to follow the differences among these models. So, I encourage the authors to summarize the similarities and differences of the processes of the schemes, according to some standards such as how to parameterize the effects of soil moisture, how to parameterize the effects of soil temperature etc. This summary will help us understanding the differences of the model and simulated results more clearly (e.g., Page 12, 349). Meanwhile, in the results section, more direct comparisons among the model simulations should be made towards the differences of processes. In addition, the comparisons were based on monthly LAI values (Page 4, line 89 and Page 9 Line 268). However, the temporal scale may cover up the real phenology characteristics. For example, based on the 8-day LAI data, Zhang et al., (2019) demonstrated that the CLM simulated growing season type is TGS in a temperate grassland, but the MODIS LAI-based type was SGS-S. It seems like that this discrepancy was not found in the study (Fig. 2b). Therefore, the monthly LAI mean output from the models may cause uncertainties on the model evaluation. Moreover, as mentioned by the authors (Page 16 Line 479), double cropping cropland can not be easily detected by the monthly LAI data, for example, a large area of winter wheat-summer maize double cropping system in the North China Plain was not detected by the method based on MODIS LAI (Fig 1 a). So, the uncertainty from the monthly LAI output from the models should be also discusses. I have no other remarks.

References: Zhang, L., Lei, H., Shen, H., Cong, Z., Yang, D., & Liu, T. (2019). Evaluating the representation of vegetation phenology in the Community Land Model 4.5 in a temperate grassland. Journal of Geophysical Research: Biogeosciences, 124(2), 187-210.

---

## Referee Comment (RC2) · Matthias Forkel (Referee) · 22 Dec 2020

Dear authors,

Many thanks for your interesting evaluation of phenology in the CRESCENDO model ensemble. Please excuse the delay in my referee report.

In my view, your study has some methodological issues (see below) that need to be resolved before I can recommend publication of the manuscript. In addition, your analysis and model-data comparisons could benefit from considering some other publications that benchmarked land surface models for phenology or that parametrized phenology

modules in DGVMs with satellite observations (Forkel et al., 2015; Jolly et al., 2005; Kelley et al., 2013; Knorr et al., 2010; MacBean et al., 2015; Schaphoff et al., 2018; Stöckli et al., 2011).

Good luck with the revisions and I'm looking forward to read a revised version.

Yours sincerely, Matthias Forkel

1 Naming of LAI datasets

The name "SENTINEL LAI" is completely misleading. Sentinels are various satellites and some of them allow to retrieve LAI (Sentinel-2 MSI and Sentinel-3 OLCI). However as far as I know, there is no operational LAI datasets from the Sentinels. The text writes that the data comes from SPOT and Proba-V and there is indeed a LAI datasets available from these sensors through the Copernicus Global Land Service. However, this dataset is harmonized from observations from the SPOT and Proba-V sensors but does not include data from any Sentinels. You need to be accurate and specific about the used dataset. If the used dataset does not include Sentinel data, it should not be named "SENTINEL". A similar issue is with the "MODIS" dataset. You need to be specific which dataset and version was used. The most recent MODIS LAI/fPAR dataset is "MOD15A2H" – is this the used dataset? Was the GIMMS LAI3g or a more recent version used?

2 Regridding of LAI and model datasets

The section about the regridding of the LAI datasets needs some more details: How were data gaps/missing values considered during regriddng? How were different land cover type considered? Did you separate the LAI of different land covers for each 0.25° grid cell?

3 Propagation of differences between datasets in the analysis

What is the reasoning behind using MODIS as the reference (L 304-306)? From your analysis, you cannot provide any evidence that the MODIS dataset would be "better"

than the other two. It would be better to propagate the differences between the satellite datasets in the entire analysis. For example, the maps could also show the agreement of the "multi-data ensemble".

4 Chapter 3.5 and Figures 6 and 7

I find the biome-averaged(?) GSS and GSE values misleading, especially for PFTs that grow on both hemispheres. How were values from both hemispheres computed? I hope you did not just use the average! Also given the fact that GSS usually delays pole-wards, I think biome-averaged GSS and GSE dates are not meaningful. Better would be to show distributions of GSS and GSE (e.g. boxplots or violin plots and separated for the northern and southern hemisphere). In addition, I wonder how differences in the spatial distribution of PFTs were considered in this data-model comparison. Was the comparison only done for grid cells where both ESA CCI and the models have the same PFT? Howe were the different PFT schemes of the models made consistent with the PFTs from the ESA CCI land cover map? If differences in PFT distribution are not considered, there might be the risk that differences in GSS and GSE are actually not related to the phenology module but to the model components that affect the spatial distribution of vegetation (e.g. establishment, mortality, disturbance). This might be especially an issue for models with dynamic (not prescribed) vegetation such as LPJ-GUESS.

Specific comments

Many sentences are long and rather difficult to read. I suggest to revise the text specifically to shorten some sentences.

L 16-19: The first sentence is very difficult to read (very long, 4 times interrupted with references). I suggest to split this into 2-3 sentences.

L 59-60: It is necessary to also shortly describe what the 4GST method actually is and how it differs from other approaches for evaluating phenology.

L 74-76: Please provide the correct link to the used dataset and not just to the Copernicus programme.

L 94-98: How were these PFTs compared to the models that might use other terminologies for PFTs?

L 156: "mostly according to the common CRESCENDO protocol": I which aspects did the model run differ from the protocol?

L 267-268: Describe how the model results were disaggregated to 0.5° resolution if their native resolution was coarser.

L 275-278: It would be helpful if a simplified version of Supp. Fig. 1 would be in the main text to immediately understand the methodology without the need to go to the supplement or previous paper.

L 283: But how was the timing of GSS and GSE defined? How did you obtain these phenophase dates from the linear regression fitted to the LAI data?

L 285-287: If the precision of the method is only 1 month, how can you report biases of 0.5 months for GSE and 0.6 months for GSS in the abstract?

L 321-323: I rather think that there is a different reason for not detecting evergreen phenology in northern forests in the LAI datasets: In the northern needle-leaf evergreen forests, tree cover is only between 40% and 60% at the used resolution of 0.5°. Hence a large part of the LAI seasonality in this regions comes either from the understory, from gaps or grasslands which indeed show a seasonality.

Figure 5: By looking at the large variability of the satellite datasets at below 40°S, I'm wondering it the same grid cells were used fro all datasets and models. How were the latitudinal gradients averaged over regions where one dataset or model shows EVG phenology (hence no GSS and GSE dates) but the others did. Was the same land/sea mask and no data mask used for all datasets and models? I assume that major differences between datasets and models originate also from the choice of the

grid cells that were included in averaging.

L 453: The maximum LAI value is also affected by the spatial resolution of the aggregated dataset because high values are increasingly averaged towards lower values.

L 481: Please avoid paragraphs that consist only of one line/sentence.

References

Forkel, M., Migliavacca, M., Thonicke, K., Reichstein, M., Schaphoff, S., Weber, U. and Carvalhais, N.: Codominant water control on global interannual variability and trends in land surface phenology and greenness, Glob Change Biol, 21(9), 3414–3435, doi:10.1111/gcb.12950, 2015.

Jolly, W. M., Nemani, R. and Running, S. W.: A generalized, bioclimatic index to predict foliar phenology in response to climate, Global Change Biology, 11(4), 619–632, doi:10.1111/j.1365-2486.2005.00930.x, 2005.

Kelley, D. I., Prentice, I. C., Harrison, S. P., Wang, H., Simard, M., Fisher, J. B. and Willis, K. O.: A comprehensive benchmarking system for evaluating global vegetation models, Biogeosciences, 10(5), 3313–3340, doi:10.5194/bg-10-3313-2013, 2013.

Knorr, W., Kaminski, T., Scholze, M., Gobron, N., Pinty, B., Giering, R. and Mathieu, P.-P.: Carbon cycle data assimilation with a generic phenology model, J. Geophys. Res., 115(G4), G04017, doi:10.1029/2009JG001119, 2010.

MacBean, N., Maignan, F., Peylin, P., Bacour, C., Bréon, F.-M. and Ciais, P.: Using satellite data to improve the leaf phenology of a global terrestrial biosphere model, Biogeosciences, 12(23), 7185–7208, doi:10.5194/bg-12-7185-2015, 2015.

Schaphoff, S., Forkel, M., Müller, C., Knauer, J., von Bloh, W., Gerten, D., Jägermeyr, J., Lucht, W., Rammig, A., Thonicke, K. and Waha, K.: LPJmL4 – a dynamic global vegetation model with managed land – Part 2: Model evaluation, Geosci. Model Dev., 11(4), 1377–1403, doi:10.5194/gmd-11-1377-2018, 2018.

[Figure]

Stöckli, R., Rutishauser, T., Baker, I., Liniger, M. a and Denning, a S.: A global reanalysis of vegetation phenology, Journal of Geophysical Research, 116(G3), G03020–G03020, doi:10.1029/2010jg001545, 2011.

---

## Author Comment (AC2) · 22 Jan 2021

See the attachment.

Please also note the supplement to this comment:
https://bg.copernicus.org/preprints/bg-2020-319/bg-2020-319-AC2-supplement.pdf

---

## Author Response (AR1)

**Replies to the comments of reviewer #1**

The study evaluated the plant phenology simulated by CRESCENDO land surface models using satellite observational LAI products. Specially, the 4GST method was applied to extract the times of start and end of growing season based on the simulated and remote sensing monthly LAI values. Then, the growing season types, variability of growing season start and end, latitudinal variability, and reginal variability were com- pared between the model simulations and satellite observations. Recommendations were also given for future model improvements. In general, the manuscript was written well, organized well, and the results were summarized clearly and interesting. So, I think the manuscript can be accepted for publication on the journal.

We thank the reviewer for her/his positive feedbacks and useful comments.

Only one main remark is that the description of the phenology schemes of the models. As we know, the phenology schemes in the models are quite different, in terms of their parameterizations of solar radiation, day-length, temperature, and soil moisture conditions. In section 2.2, the description of phenology schemes makes me a little bit hard to follow the differences among these models. So, I encourage the authors to summarize the similarities and differences of the processes of the schemes, according to some standards such as how to parameterize the effects of soil moisture, how to parameterize the effects of soil temperature etc. This summary will help us understanding the differences of the model and simulated results more clearly (e.g., Page 12, 349).

We thank the reviewer. In the revised version of the manuscript we summarize and re-organize the section 2.2 in order to make the phenology scheme description of each land surface model more concise and easier to compare. Besides, we update table 1 to contain further details on the different phenology parameterization adding information on root zone depth, temperature and moisture thresholds, and variables used in detecting the phenology phases:

**Table 1.** Grid spatial resolution used for each land surface model (LSM) and brief summary of their main features. PFT stands for Plant Functional Type and CFT stands for Crop Functional Type. The second part gives further details on the phenology schemes.

| LSM | Original Resolution | PFT | Soil level | CFT | Phenology scheme | Phenology drivers | Root zone |
|---|---|---|---|---|---|---|---|
| CLM 4.5 | 1.25° x 0.9375° | 15 | 15 | 1 C3 | evergreen; seasonal-deciduous; stress-deciduous | Soil temperature; soil moisture; day-length | Zeng (2001) |
| CLM 5.0 | 0.5° x 0.5° | 16 | 20 | 2 C3 | evergreen; seasonal-deciduous; stress-deciduous | Soil temperature; moisture day-length; precipitation | Jackson et al. (1996) |
| JULES-ES | 1.875° x 1.25° | 13 | 4 | 1 C3, 1 C4 | Deciduous trees | Surface temperature | Wiltshire et al. (2020a) |
| JSBACH | 1.9° x 1.9° | 12 | 5 | 1 C3, 1 C4 | evergreen; summergreen; raingreen; grasses; tropical crops; extra-tropical crops | air temperature; soil temperature; soil moisture; NPP | Kleidon (2004) |
| LPJ-GUESS | 0.5° x 0.5° | 25 | 2 | 3 C3, 2 C4 | evergreen; seasonal-deciduous; stress-deciduous | Soil temperature; soil moisture | Root in top soil layer[†]: Herbaceous PFTs 90%, Woody PFTs 60% |
| ORCHIDEE | 0.5° x 0.5° | 15 | 11 | 1 C3, 1 C4 | deciduous; dry and semi-arid; grasses and crops; grasses and crops | Air temperature; soil moisture; soil moisture | Exponential profile within the 2m soil; Krinner et al. (2005) |
| ISBA-CTRIP | 1° x 1° | 16 | 14 | 1 C3, 1 C4 | leaf biomass | Leaf biomass | Canadell et al. (1996) |

| LSM | Temperature variable | Temperature threshold | Moisture variable | Moisture Threshold | Reference |
|---|---|---|---|---|---|
| CLM 4.5 | Third soil layer* temperature | 0 °C | Third soil layer* water potential | -2 MPa | Oleson et al. (2013) |
| CLM 5.0 | Third soil layer** temperature | 0 °C | Third soil layer** water potential | -0.6 MPa | Lawrence et al. (2018) |
| JULES-ES | mean daily surface temperature | 5 °C | None | N/A | Clark et al. (2011) |
| JSBACH | depending on the phenology: air, pseudo-soil temperature | depending on the phenology: 4°C, 10°C | soil moisture in the root zone | wilting point of 0.35 m/m | Mauritsen et al. (2019) |
| LPJ-GUESS | Top soil layer[†] temperature | 5 °C | water stress scalar ($\omega$) | minimum of $\omega$ ($\omega_{min}$) | Smith et al. (2014) |
| ORCHIDEE | mean daily air temperature | Sum above -5 °C | soil moisture in root zone | Increase of moisture over 5 days | Botta et al. (2000) |
| ISBA-CTRIP | no phenology model: LAI is deduced from leaf biomass through Specific Leaf Area, which depends on nitrogen content | | | | Delire et al. (2020) |

* about 6cm depth. ** 9cm depth. [†] 25cm depth.

Meanwhile, in the results section, more direct comparisons among the model simulations should be made towards the differences of processes.

We increase the discussion and reference to the difference among models in the result sections of the revised version of the manuscript, especially in sections 3.2, 3.3, 3.4, and 3.5.

In addition, the comparisons were based on monthly LAI values (Page 4, line 89 and Page 9 Line 268). However, the temporal scale may cover up the real phenology characteristics. For example, based on the 8-day LAI data, Zhang et al., (2019) demonstrated that the CLM simulated growing season type is TGS in a temperate grassland, but the MODIS LAI-based type was SGS-S. It seems like that this discrepancy was not found in the study (Fig. 2b). There- fore, the monthly LAI mean output from the models may cause uncertainties on the model evaluation. Moreover, as mentioned by the authors (Page 16 Line 479), double cropping cropland can not be easily detected by the monthly LAI data, for example, a large area of winter wheat-summer maize double cropping system in the North China Plain was not detected by the method based on MODIS LAI (Fig 1 a). So, the uncertainty from the monthly LAI output from the models should be also discusses. I have no other remarks.

We thank the reviewer.
The use of monthly data is another limitation in our methodology, indeed.
Following the reviewer's suggestion, we add this point to the discussion in Section 4.3:

*"Another limitation of the present evaluation is the monthly temporal frequency. Data at a higher frequency, indeed, might lead to a more detailed bias assessment. The use of a different temporal frequency may also influence phenology type detection. For example, Peano et al. (2019), that uses 15-day LAI data, detect a slightly different distribution of CLM4.5 SGS-D and TGS types in Australia, Horn of Africa, and Brazil. Similarly, Zhang et al. (2019), which analyses CLM4.5 in Northeast China with 8-day LAI data, obtain TGS type in areas recognized as SGS-S in the present analysis."*

**Replies to the comments of reviewer #2**

In my view, your study has some methodological issues (see below) that need to be resolved before I can recommend publication of the manuscript. In addition, your analysis and model-data comparisons could benefit from considering some other publications that benchmarked land surface models for phenology or that parametrized phenology modules in DGVMs with satellite observations (Forkel et al., 2015; Jolly et al., 2005; Kelley et al., 2013; Knorr et al., 2010; MacBean et al., 2015; Schaphoff et al., 2018; Stöckli et al., 2011).

We thank the reviewer for the feedbacks and useful comments, which are addressed below. We take into consideration the suggested publications in the revision of our manuscript.

1 Naming of LAI datasets

The name "SENTINEL LAI" is completely misleading. Sentinels are various satellites and some of them allow to retrieve LAI (Sentinel-2 MSI and Sentinel-3 OLCI). However as far as I know, there is no operational LAI datasets from the Sentinels. The text writes that the data comes from SPOT and Proba-V and there is indeed a LAI datasets available from these sensors through the Copernicus Global Land Service. However, this dataset is harmonized from observations from the SPOT and Proba-V sensors but does not include data from any Sentinels. You need to be accurate and specific about the used dataset. If the used dataset does not include Sentinel data, it should not be named "SENTINEL". A similar issue is with the "MODIS" dataset. You need to be specific which dataset and version was used. The most recent MODIS LAI/fPAR dataset is "MOD15A2H" – is this the used dataset? Was the GIMMS LAI3g or a more recent version used?

We thank the reviewer pointing out these inaccuracies. The naming "SENTINEL" derives from the initial source of Copernicus LAI data. This choice switched to SPOT and Proba-V data, but the naming remained unchanged. We rename SENTINEL as Copernicus Global Land Service (CGLS) in the revised version of the manuscript.

About MODIS data, they are based on the MOD15A2H and MYD15A2H version 6 database. The LAI3g data is the GIMMS version. We add this information explicitly in the revised version of the manuscript.

2 Regridding of LAI and model datasets

The section about the regridding of the LAI datasets needs some more details: How were data gaps/missing values considered during regriddng? How were different land cover type considered? Did you separate the LAI of different land covers for each 0.25° grid cell?

The three satellite products used in the present analysis are gap-filled data. In particular, Copernicus Global Land Service data applies a linear interpolation in a local moving window of 128-day length to fill gaps, as described by Verger et al (2011). LAI3g data use spline interpolation and seasonal profile methodologies as done for NDVI3g data. MODIS data use a combination of Terra and Aqua data, a temporal averaging, and climatology to fill gaps. For this reason, no specific gaps/missing values consideration is required during the regridding operations. Besides, differences in land cover type have not been taken into account in the regridding operations. The revised version of the paper describes these points:

*"The 2000-2011 period is common to the three satellite datasets and it is used in the present analysis. The satellite observations are aggregated into monthly values and regridded, by means of a first order conservative remapping scheme (Jones, 1999), to a regular 0.5° x 0.5° grid for consistency with the LSMs' output. The regridding operation is directly applied to the gap-filled satellite data. Note that regridding does not employ any specific treatment for differences in land cover."*

**3 Propagation of differences between datasets in the analysis**

What is the reasoning behind using MODIS as the reference (L 304-306)? From your analysis, you cannot provide any evidence that the MODIS dataset would be "better" than the other two. It would be better to propagate the differences between the satellite datasets in the entire analysis. For example, the maps could also show the agreement of the "multi-data ensemble".

We agree with the reviewer on the absence of any evidence to say that MODIS is better than the other satellite data. In fact, we are not saying that MODIS is better than the other observations. However, graphical constraints impose us to choose one observational dataset to be used as a reference. As suggested by the reviewer, we updated Figures 2, 3, and 4 to keep track of the agreement among satellite data despite the choice of MODIS as reference. This is also described in the manuscript text as follow:

*"Keeping these differences in mind, the MODIS data are used as a graphical reference in Figures 3, 4, and 5. These figures keep track of the agreement among satellite data despite the choice of MODIS as reference. Figures using CGLS and LAI3g as a graphical reference are presented in the supplementary material."*

[Figure]

Updated Figure 2 -> Figure 3: Global climatological (averaged over 2000-2011) distribution of the four main growing season modes for (a) MME; (b) CLM 4.5; (c) CLM 5.0; (d) JULES-ES; (e) JSBACH; (f) LPJ-GUESS; (g) ORCHIDEE; (h) ISBA-CTRIP. The areas characterized by the same type of LSMs and MODIS (Figure 2a) are coloured. These common areas are called agreement regions. Index values: (purple) evergreen; (green) single season with summer LAI peak; (cyan) single growing season with summer dormancy; (orange) two growing seasons type. White regions are for disagreement areas. Above this selection, areas of agreement between satellite products are shaded with a different hatching pattern:

MODIS and LAI3g (Figure 2d) slash hatching (/); MODIS and CGLS (Figure 2g) backslash hatching (\);
MODIS, CGLS, and LAI3g crossed hatching (X).

[Figure]

Update Figure 3 -> Figure 4: Global climatological (averaged over 2000-2011) differences in growing
season start timings (GSS) between (a) Multi-Model Ensemble mean (MME); (b) CLM 4.5; (c) CLM 5.0;

(d) JULES-ES; (e) JSBACH; (f) LPJ-GUESS; (g) ORCHIDEE; (h) ISBA-CTRIP and MODIS (Figure 2b). The green regions represent areas of agreement between MODIS and LSMs. Yellow-red colors correspond to areas where models timings are later compared to MODIS, while blue-violet colors correspond to areas where models timings are earlier compared to MODIS. Regions where GSS timings are not computed, such as non-vegetated and evergreen areas, are in white. Above this selection, areas of agreement between satellite products are shaded with a different hatching pattern: MODIS and LAI3g (Figure 2d) slash hatching (/); MODIS and CGLS (Figure 2g) backslash hatching (\); MODIS, CGLS, and LAI3g crossed hatching (X). Note that the GSS in the TGS regions corresponds to the GSS of the first growing season cycle.

[Figure]

Updated Figure 4 -> Figure 5: As Figure 4 but for growing season end (GSE) timings. Note that the GSE in the TGS regions corresponds to the GSE of the second growing season cycle.

*I find the biome-averaged(?) GSS and GSE values misleading, especially for PFTs that grow on both hemispheres. How were values from both hemispheres computed? I hope you did not just use the average! Also given the fact that GSS usually delays pole- wards, I think biome-averaged GSS and GSE dates are not meaningful. Better would be to show distributions of GSS and GSE (e.g. boxplots or violin plots and separated for the northern and southern hemisphere). In addition, I wonder how differences in the spatial distribution of PFTs were considered in this data-model comparison. Was the comparison only done for grid cells where both ESA CCI and the models have the same PFT? Howe were the different PFT schemes of the models made consistent with the PFTs from the ESA CCI land cover map? If differences in PFT distribution are not considered, there might be the risk that differences in GSS and GSE are actually not related to the phenology module but to the model components that affect the spatial distribution of vegetation (e.g. establishment, mortality, disturbance). This might be especially an issue for models with dynamic (not prescribed) vegetation such as LPJ- GUESS.*

We thank the reviewer for these suggestions. In the revised version of the manuscript: (1) we distinct between hemispheres by separating the biomes that are present in both hemispheres; (2) we update figures 6 and 7 into boxplots.

The biome-level analysis was performed based on the regional map retrieved from the ESA-CCI data. The same regional distinction, then, is applied to all the evaluated models. This choice allows for a standard distribution among models. We need to keep in mind that differences among models can derive from phenology scheme and differences in cover map. For example, the phenology of LPJ-GUESS will change both due to changes in phenology and vegetation mixture. But this is a feature of LPJ-GUESS that should be taken into account in our assessment. This point is made clear in the revised version of the manuscript.

Chapter 3.5 and Figures 6 and 7 are revised as follow:

*"3.5 Regional Variability*

[revised manuscript text omitted]

Updated Figure 7 -> Figure 8: as Figure 7 but for the growing season end (GSE) timings. In this case, the winter season is central along the y-axis.

**Specific comments**

Many sentences are long and rather difficult to read. I suggest to revise the text specifically to shorten some sentences.

We thank the reviewer. We update the text accordingly.

L 16-19: The first sentence is very difficult to read (very long, 4 times interrupted with references). I suggest to split this into 2-3 sentences.

We thank the reviewer for this suggestion. This sentence has been rephrased as:

*"Plant phenology and its variability have a substantial influence on the terrestrial ecosystem (e.g. Churkina et al., 2005;Kuchariket al., 2006;Berdanier and Klein, 2011) and land-atmosphere interactions (e.g. Cleland et al., 2007;Richardson et al., 2013; Keenan et al., 2014). Moreover,*

*recent decades observations show modifications in both spring and autumn phenology under global warming (e.g. Menzel et al., 2006; Richardson et al., 2013; Park et al., 2016; Zhu et al., 2016; Chen et al., 2020; Zhang et al., 2020). For these reasons, phenology variability is one of the indicators of climate change (e.g. Schwartz et al., 2006; Soudani et al., 2008; Jeong et al., 2011)."*

L 59-60: It is necessary to also shortly describe what the 4GST method actually is and how it differs from other approaches for evaluating phenology.

We agree with the reviewer and update this paragraph as follow:

*"Vegetation phenology can be assessed by considering different plant features and indices, such as leaves colour (e.g. normalized difference vegetation index, NDVI, Churkina et al., 2005; Keenan et al., 2014), the fraction of absorbed photosynthetically active radiation (e.g. Kelley et al., 2013; Forkel et al., 2015), or canopy density (e.g. LAI Murray-Tortarolo et al., 2013; Peano et al., 2019). Each methodology presents skills and limitations (e.g Forkel et al., 2015). In this work, the novel Four Growing Season Types (4GST) methodology developed by Peano et al. (2019) is used to evaluate phenology. This method proved good skill in capturing the principal global phenology cycles (Peano et al., 2019), and integrates a broader spectrum of growing season modes compared to previous techniques (e.g. Murray-Tortarolo et al., 2013). The set of growing season modes investigated in 4GST are (1) evergreen phenology; (2) single growing season with summer LAI peak; (3) single growing season with summer dormancy; and (4) two growing seasons. 4GST uses LAI data to evaluate phenology. Most ecosystem and climate models introduce 'leaf area' as a fundamental state parameter describing the interactions between the biosphere and the atmosphere. The most common measure of the area of leaves is the LAI, which is generally defined as the one-sided leaf surface area divided by the ground area in $m^2/m^2$ (Chen and Black, 1992). In addition, LAI is the key variable by which LSMs scale-up leaf-level processes to canopy and ecosystem scale exchanges of carbon, energy, and water. This makes the LAI a reasonable choice for the evaluation of the LSMs' phenology (Murray-Tortarolo et al., 2013; Peano et al., 2019)."*

L 74-76: Please provide the correct link to the used dataset and not just to the Copernicus programme.

We update the link to the proper Copernicus page:

https://land.copernicus.eu/global/products/LAI

L 94-98: How were these PFTs compared to the models that might use other terminologies for PFTs?

The biome-level analysis is performed based on the regional map retrieved from the ESA-CCI data. The same regional distinction, then, is applied to all the evaluated models. This choice allows for a standard distribution among models. We need to keep in mind that differences among models can derive from phenology scheme and differences in cover map. This point is made explicit in the revised version of the manuscript. To make this choice clear in the methodology section, we rephrase this sentence as follow:

*"To perform biome-level analysis, the observed ESA CCI land cover map (https://www.esa-landcover-cci.org/) has been used to define a standard regional vegetation distribution."*

L 156: "mostly according to the common CRESCENDO protocol": I which aspects did the model run differ from the protocol?

We rephrase the sentence to make the difference clearer:

*"For the simulations described here, JSBACH3.2 is run offline at T63 (~1.9°) resolution. Simulations were conducted without natural changes in the land cover, instead, a static map of natural land cover based on Pongratz et al.(2008) was used. Anthropogenic land cover changes were applied using land-use transitions (see Reick et al., 2013) derived from the LUH2 forcing, whereby rangelands were treated as natural vegetation (see also Mauritsen et al.,2019). Simulations were conducted according to the common CRESCENDO protocol as described in section 2.3, with the only difference that land-use change was already simulated starting 1700 to avoid a cold start problem when applying land-use transitions."*

L 267-268: Describe how the model results were disaggregated to 0.5° resolution if their native resolution was coarser.

We add this information in the revised version of the manuscript as follow:

*"CLM4.5, JULES-ES, JSBACH, and ISBS-CTRIP perform their simulations at a coarser resolution. Their output are regridded by applying a first-order conservative remapping method (Jones, 1999)."*

L 275-278: It would be helpful if a simplified version of Supp. Fig. 1 would be in the main text to immediately understand the methodology without the need to go to the supplement or previous paper.

We agree with the reviewer and we move a modified version of Supplementary Figure 1 into the main text. This is the new Figure 1 in the revised version of the manuscript. Note that this changes the figure numbering in the revised version of the manuscript.

[Figure]

Figure 1: scheme of the Four Growing Season Type method used in evaluating start and end of the growing season.

L 283: But how was the timing of GSS and GSE defined? How did you obtain these phenophase dates from the linear regression fitted to the LAI data?

The four linear regressions are used to distinguish the phenology type in each grid point. Once the phenology type is defined, the start and end of the growing season are detected when 20% of the LAI annual variability is reached. In case of the two growing seasons type, the critical threshold is defined twice: one for each growing season cycle. Areas featuring annual LAI changes smaller than 25% of the mean LAI are defined as evergreen regions. This information is summarized in the revised version of the manuscript as follow:

*"The 4GST method allows to evaluate start and end of the growing season and the global spatial distribution of four main growing season types: (1) evergreen (EVG); (2) single growing season peaking in summer (SGS-S); (3) single growing season with summer dormancy (SGS-D); (4) two growing seasons (TGS). The EVG type is identified when relative changes in LAI annual cycle are smaller than 25% of local LAI mean value. Note that GSS and GSE timings are not detected in EVG areas. The other three types are distinguished based on LAI slopes and transition timings*

*as illustrated and summarized in Figure 1. When one single growing season is identified, SGS-S and SGS-D are discerned based on the peak-month (i.e. in the Northern Hemisphere (NH) SGS-S is detected when LAI peak occurs between April and September, otherwise, SGS-D is detected, vice versa in the Southern Hemisphere (SH)). The timings of start and end of the growing season are identified through a critical threshold set to 20% of the annual LAI cycle (Figure 1). TGS, instead, is identified when two growing seasons at least three-month-long are detected and GSS and GSE timings are identified for each cycle. Further details can be found in Peano et al. (2019)."*

L 285-287: If the precision of the method is only 1 month, how can you report biases of 0.5 months for GSE and 0.6 months for GSS in the abstract?

We think this phrase is misleading in this part of the manuscript. For this reason, we delete it in the revised version and add this point in the discussion section of the revised version of the manuscript:

*"Another limitation of the present evaluation is the monthly temporal frequency. Data at a higher frequency, indeed, might lead to a more detailed bias assessment. The use of a different temporal frequency may also influence phenology type detection. For example, Peano et al. (2019), that uses 15-day LAI data, detect a slightly different distribution of CLM4.5 SGS-D and TGS types in Australia, Horn of Africa, and Brazil. Similarly, Zhang et al. (2019), which analyses CLM4.5 in Northeast China with 8-day LAI data, obtain TGS type in areas recognized as SGS-S in the present analysis."*

L 321-323: I rather think that there is a different reason for not detecting evergreen phenology in northern forests in the LAI datasets: In the northern needle-leaf evergreen forests, tree cover is only between 40% and 60% at the used resolution of 0.5°. Hence a large part of the LAI seasonality in this regions comes either from the understory, from gaps or grasslands which indeed show a seasonality.

We thank the reviewer mentioning this mechanism. The gridded LAI data accounts for the LAI variability of the set of PFTs available in the grid-cell. For this reason, the understory in these areas may influence the grid-level LAI seasonality. We add this mechanism in the revised version of the manuscript as follow:

*"It is noteworthy that the evergreen type is correctly detected in the broad-leaf evergreen tropical areas in both satellite observations and LSMs (Figure 3). On the contrary, the high-latitude needle-leaf evergreen regions are partially represented in LSMs, while satellite data do not catch these areas due to satellite limitations resulting from the impact of cloud and snow cover on light availability during the winter season (see Section 4.2). Besides, the variability of understory and secondary PFTs may influence LAI seasonality representation."*

Figure 5: By looking at the large variability of the satellite datasets at below 40°S, I'm wondering it the same grid cells were used fro all datasets and models. How were the latitudinal gradients averaged over regions where one dataset or model shows EVG phenology (hence no GSS and GSE dates) but the others did. Was the same land/sea mask and no data mask used for all datasets and

models? I assume that major differences between datasets and models originate also from the choice of the grid cells that were included in averaging.

We thank the reviewer pointing out this aspect of the latitudinal evaluation. The large variability in the region below 40°S may be linked to the reduced amount of vegetated area. When the comparison is made on a few grid-points, then, the impact of EVG grid-points is more evident. For this reason, we describe this mechanism in the revised version of the manuscript:

*"Large variability is spotted in the region below 40°S. The reduced amount of vegetated land area may cause this behaviour. A different growing season type detection in this area, such as a different size of the evergreen region (Figure 2), may, indeed, extensively influence the GSS and GSE detection, which is the case for the satellite products (Figure 6), especially LAI3g."*

L 453: The maximum LAI value is also affected by the spatial resolution of the aggregated dataset because high values are increasingly averaged towards lower values.

We thanks the reviewer to point out the limit given by the regridding procedure. We revise the text as follow:

*"LAI satellite data are also affected by the applied regridding and gap-filling algorithms, which could create spurious seasonal cycles as well as smooth the observed phenology season (e.g. Kandasamy et al., 2013; Chen et al., 2017). In addition, the observed reflectance saturates in regions characterized by dense canopies reaching prescribed LAI upper limits (e.g. 7.0 $m^2/m^2$ in MODIS and LAI3g; Myneni et al., 2002; Maignan et al., 2011)."*

L 481: Please avoid paragraphs that consist only of one line/sentence.

We thanks the reviewer for this suggestion. We change the text accordingly.

---

## Author Response (AR2)

The authors revised the manuscript carefully, I'm generally satisfied with the revisions and improvements. I have only one minor remark. The drivers of leaf onset and leaf senescence may be different. In table 1, the authors seem to mix the drivers and threshold values of the two processes? Or the authors only listed the drivers of leaf onset? Please clarify it. It's better to list the drivers and threshold values separately.

We thank the reviewer for appreciating our revisions.
Table 1 contains two sections: a first part describing the main features of each land surface model, and a second part detailing the temperature and moisture thresholds for the start and end of the growing season in the phenology schemes. In the revised version of table 1, we report this difference clearly in the table caption and we distinguish between thresholds for the start and end of the growing season.

**Table 1.** Grid spatial resolution used for each land surface model (LSM) and brief summary of their main features. PFT stands for Plant Functional Type and CFT stands for Crop Functional Type. The second part gives further details on the temperature and moisture thresholds for start (S) and end (E) of the growing season in the phenology schemes.

| LSM | Original Resolution | PFT | Soil level | CFT | Phenology scheme | Phenology drivers | Root zone |
|---|---|---|---|---|---|---|---|
| CLM 4.5 | 1.25° x 0.9375° | 15 | 15 | 1 C3 | evergreen; seasonal-deciduous; stress-deciduous | Soil temperature; soil moisture; day-length | Zeng (2001) |
| CLM 5.0 | 0.5° x 0.5° | 15 | 20 | 2 C3 | evergreen; seasonal-deciduous; stress-deciduous | Soil temperature; moisture day-length; precipitation | Jackson et al. (1996) |
| JULES-ES | 1.875° x 1.25° | 13 | 4 | 1 C3, 1 C4 | Deciduous trees | Surface temperature | Wiltshire et al. (2020a) |
| JSBACH | 1.9° x 1.9° | 12 | 5 | 1 C3, 1 C4 | evergreen; summergreen; raingreen; grasses; tropical crops; extra-tropical crops | air temperature; soil temperature; soil moisture; NPP | Kleidon (2004) |
| LPJ-GUESS | 0.5° x 0.5° | 25 | 2 | 3 C3, 2 C4 | evergreen; seasonal-deciduous; stress-deciduous | Air temperature; soil moisture | Root in top soil layer[†]: Herbaceous PFTs 90%, Woody PFTs 60% — Krinner et al. (2005) |
| ORCHIDEE | 0.5° x 0.5° | 15 | 11 | 1 C3, 1 C4 | deciduous; dry and semi-arid; grasses and crops | Air temperature; soil moisture; soil moisture | Exponential profile within the 2m soil — Krinner et al. (2005) |
| ISBA-CTRIP | 1° x 1° | 16 | 14 | 1 C3, 1 C4 | leaf biomass | Leaf biomass | Canadell et al. (1996) |

| LSM | Temperature variable | Temperature threshold | Moisture variable | Moisture Threshold | Reference |
|---|---|---|---|---|---|
| CLM 4.5 | Third soil layer* temperature | 0 °C (S,E) | Third soil layer* water potential | -2 MPa (S,E) | Oleson et al. (2013) |
| CLM 5.0 | Third soil layer** temperature | 0 °C (S,E) | Third soil layer** water potential | -0.6 MPa (S), -2 MPa (E) | Lawrence et al. (2018) |
| JULES-ES | mean daily surface temperature | depending on PFT and whether S/E: 5 °C, 6.85 °C (S,E) | None | N/A | Clark et al. (2011) |
| JSBACH | depending on the phenology: air; pseudo-soil temperature | depending on the phenology and whether S/E: 4°C, 10°C, or critical heat sum (S,E) | soil moisture in the root zone | wilting point of 0.35 m/m (S,E)[†] | Mauritsen et al. (2019) / Reick et al. (2021) |
| LPJ-GUESS | Mean daily air temperature | Sum above 5 °C (S) | water stress scalar ($\omega$) | minimum of $\omega$ ($\omega_{min}$) (S,E) | Smith et al. (2014) |
| ORCHIDEE | mean daily air temperature weekly temperature | Sum above -5 °C, 0°C (S) depending on PFT: 2, 5, 7, 10, 12 °C (E) | weekly relative soil moisture | 5 days moisture increase (S) depending on PFT: 0.2, 0.3 $m^3/m^3$ (E) | Botta et al. (2000) / Krinner et al. (2005) |
| ISBA-CTRIP | no phenology model: LAI is deduced from leaf biomass through Specific Leaf Area, which depends on nitrogen content | | | | Delire et al. (2020) |

* about 6cm depth. ** 9cm depth. [†] relative soil moisture content above which growth is possible and below which growth stops and shedding sets in.